# ComGS: Efficient 3D Object-Scene Composition via Surface Octahedral Probes

Jian Gao[1]*   Mengqi Yuan[1]*   Yifei Zeng[1]   Chang Zeng[1]   Zhihao Li[2]   Zhenyu Chen[2]

Weichao Qiu[2]   Xiao-Xiao Long[1]   Hao Zhu[1]   Xun Cao[1]   Yao Yao[1][✉]

[1]Nanjing University   [2]Huawei Noah's Ark Lab

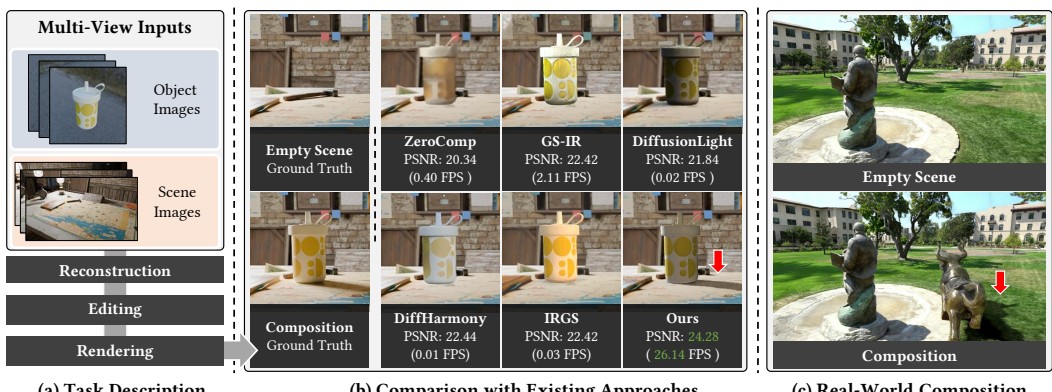

Figure 1: **(a) Task:** From multi-view images, we aim for *realistic 3D object–scene composition* via reconstruction, editing, and rendering. **(b) Comparison:** Compared to existing methods, our approach achieves superior *harmonious appearance* and *plausible shadows* (red arrows), while running at around 26 FPS. **(c) Application:** Our method also performs well on *real-world* captures.

## ABSTRACT

Gaussian Splatting (GS) enables immersive rendering, but realistic 3D object–scene composition remains challenging. Baked appearance and shadow information in GS radiance fields cause inconsistencies when combining objects and scenes. Addressing this requires relightable object reconstruction and scene lighting estimation. For relightable object reconstruction, existing Gaussian-based inverse rendering methods often rely on ray tracing, leading to low efficiency. We introduce Surface Octahedral Probes (SOPs), which store lighting and occlusion information and allow efficient 3D querying via interpolation, avoiding expensive ray tracing. SOPs provide at least a $2\times$ speedup in reconstruction and enable real-time shadow computation in Gaussian scenes. For lighting estimation, existing Gaussian-based inverse rendering methods struggle to model intricate light transport and often fail in complex scenes, while learning-based methods predict lighting from a single image and are viewpoint-sensitive. We observe that 3D object–scene composition primarily concerns the object's appearance and nearby shadows. Thus, we simplify the challenging task of full scene lighting estimation by focusing on the environment lighting at the object's placement. Specifically, we capture a 360° reconstructed radiance field of the scene at the location and fine-tune a diffusion model to complete the lighting. Building on these advances, we propose ComGS, a novel 3D object–scene composition framework. Our method achieves high-quality, real-time rendering at around 26 FPS, produces *visually harmonious* results with *vivid shadows*, and requires only 36 seconds for editing. The code and dataset are available at https://nju-3dv.github.io/projects/ComGS/.

---

*Equally Contributed.

## 1 INTRODUCTION

Gaussian Splatting (GS) (Kerbl et al., 2023) has emerged as a powerful point-based differentiable rendering technique, enabling high-fidelity 3D reconstruction and rendering from multi-view images. Despite its success, realistic 3D object–scene composition is still challenging, as the GS radiance field inherently bakes appearance and shadows. Realistic object-scene composition requires integrating objects into scenes while ensuring **visual harmony** and physically **plausible shadows**. Two critical obstacles must be addressed to achieve this goal: (1) *relightable object reconstruction* to account for appearance variations, and (2) *scene lighting estimation* to enable object relighting and realistic shadows.

For relightable object reconstruction, existing Gaussian-based inverse rendering approaches (Gu et al., 2025; Sun et al., 2025) often rely on costly ray tracing for occlusion and indirect lighting, leading to low efficiency. This constitutes a significant bottleneck for real-time object-scene composition. Meanwhile, lighting estimation in complex scenes remains an open problem. Gaussian-based inverse rendering (Liang et al., 2024b) methods struggle due to intricate light transport in complex scenes, while learning-based methods (Zhan et al., 2021; Phongthawee et al., 2024), which typically predict lighting from a single image, fail to ensure multi-view consistency. Consequently, both categories of methods fall short in providing reliable lighting for realistic object–scene composition.

To address these challenges, we propose two key innovations. First, we introduce *Surface Octahedral Probes (SOPs)* for efficient relightable Gaussian object reconstruction. SOPs store indirect lighting and occlusion, allowing shading points to access this information via interpolation rather than costly ray tracing. Thanks to the efficiency of SOPs, our method achieves at least a *2× speedup* in reconstruction compared to state-of-the-art approaches, while maintaining comparable accuracy.

Second, we observe that achieving visually harmonious and plausible shadows in object–scene composition does not require perfectly decoupling scene lighting, which presents significant technical challenges. In practice, it is sufficient to estimate the lighting around the object. Accordingly, we reformulate scene lighting estimation as the task of inferring an environment map from partially reconstructed Gaussian radiance fields. We first perform a 360-degree trace of the Gaussian scene to extract partial radiance and then input it into a fine-tuned Diffusion Model (Rombach et al., 2022) to generate a complete lighting estimation. Our approach leverages the existing Gaussian radiance field, leading to more accurate and realistic lighting results.

Building on these advances, we present ComGS, a novel object-scene composition framework leveraging SOPs. Our solution operates in three stages: (1) *Reconstruction*, for relightable object and scene reconstruction; (2) *Editing*, including lighting estimation and SOPs-based occlusion caching; and (3) *Rendering*, covering object relighting and shadow casting. By combining reconstructed relightable objects with estimated lighting, we achieve visual coherence and vivid shadows. SOPs cache occlusion during the editing stage, enabling efficient calculation of plausible shadows during rendering. For static object placements, our method achieves approximately 26 FPS rendering, with an editing time of 36 seconds. We summarize our key contributions as follows:

- A complete 3D object–scene composition framework that ensures visual harmony and plausible shadows, achieving 26 FPS rendering with an editing time of only $36\,\mathrm{s}$.
- A novel inverse rendering pipeline using Surface Octahedral Probes (SOPs) for efficient relightable object reconstruction, providing over 2× speedup compared to SOTA methods.
- Extensive evaluations on SynCom, public datasets, and phone captures that show $+1.4\,\mathrm{dB}$ PSNR, 21% higher 3D consistency, and 56% greater harmony over existing methods, highlighting our framework's potential for immersive 3D applications.

## 2 RELATED WORK

**Object-Scene Composition**  Most methods aim to combine foreground object images with background images. Some approaches (Niu et al., 2023; Zhou et al., 2024) focus specifically on achieving visual harmony between foreground and background. Diffusion-based methods (Chen et al., 2024; Zeng et al., 2024; Liang et al., 2024a) leverage generative models for flexible object placement and harmonious appearance. Intrinsics- and physics-based methods (Careaga et al., 2023; Zhang et al.,

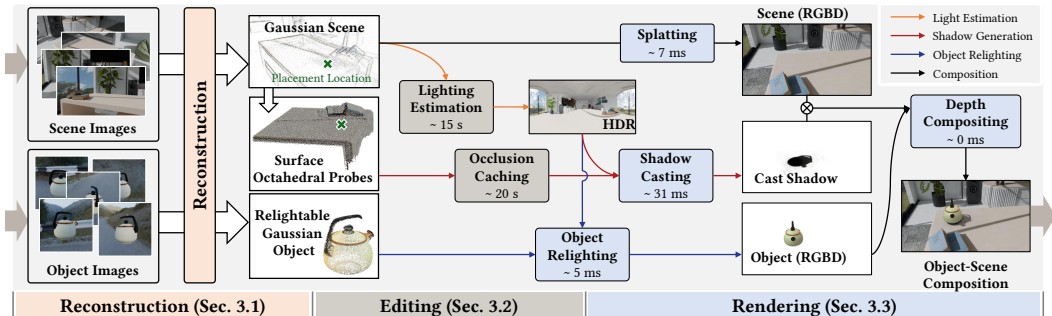

Figure 2: **Realistic 3D Object–Scene Composition Pipeline.** Our approach consists of 3 stages: *reconstruction* (Sec. 3.1), where we reconstruct the Gaussian scene and relightable Gaussian object from multi-view images; *editing* (Sec. 3.2), where we estimate scene lighting and cache occlusion using Surface Octahedral Probes; and *rendering* (Sec. 3.3), where we perform splatting, object relighting, shadow casting, and depth compositing. The pipeline achieves visually harmonious results with realistic shadows and near-real-time performance.

2025) decompose scene images into physical properties to enable photorealistic integration. Some works extend to 3D object insertion (Tarko et al., 2019; Ye et al., 2024b; Jin et al., 2025), using mesh-based representations or NeRF (Mildenhall et al., 2020), but they often rely on simplified assumptions or focus mainly on appearance, leaving effects such as shadows and complex lighting largely unhandled. MV-CoLight (Ren et al., 2025) accomplishes object composition with a feed-forward architecture, while its performance suffers from the domain gap of the training data.

**Inverse Rendering**   Inverse rendering approaches built on mesh (Hasselgren et al., 2022; Dai et al., 2025) and NeRF (Zhang et al., 2021a;b; 2022; Yao et al., 2022; Jin et al., 2023) have proven successful. Recently, Gaussian Splatting (GS) (Kerbl et al., 2023) has enabled more efficient inverse rendering (Jiang et al., 2024; Gao et al., 2024; Gu et al., 2025; Huang et al., 2025) thanks to its fast rendering. However, inverse rendering still struggles with complex scenes due to incomplete scene captures and challenging light transport. While some GS-based methods (Liang et al., 2024b; Chen et al., 2025) attempt scene-level reconstruction, they often rely on simplified lighting assumptions, resulting in inaccurate lighting decomposition and reduced realism in object-scene composition.

**Lighting Estimation**   Early methods (Gardner et al., 2017; Wang et al., 2022a; Zhan et al., 2021; Somanath & Kurz, 2021) estimate high dynamic range (HDR) environment maps from a single low dynamic range (LDR) photo to relight virtual objects for scene blending. For spatially-varying lighting, some methods predict per-pixel illumination (Li et al., 2020) or use 3D octree-based representations (Wang et al., 2024). Recently, DiffusionLight (Phongthawee et al., 2024) leverages pretrained diffusion models for improved generalization. However, relying on a single image often leads to multi-view inconsistencies. While Lyu et al. (2023) integrate inverse rendering into diffusion denoising for 360° illumination, their method relies on an explicit Mesh-based representation.

## 3   METHOD

We propose a realistic 3D object-scene composition pipeline, as shown in Figure 2. The pipeline consists of three stages: reconstruction, editing, and rendering. Each stage is detailed as follows.

### 3.1   RECONSTRUCTION

Our reconstruction process is divided into two steps. For objects, we apply both steps to reconstruct a relightable Gaussian object. For scenes, we perform only the first step for radiance field.

**Multi-Target Rendering**   To achieve better geometry than 3DGS (Kerbl et al., 2023), 2D Gaussian Splatting (2DGS) (Huang et al., 2024) employs surfels in 3D space as rendering primitives. Each surfel is defined by center $\mathbf{p}_i$, quaternion $\mathbf{q}_i$ for orientation, and scaling vector $\mathbf{s}_i$ for deformation.

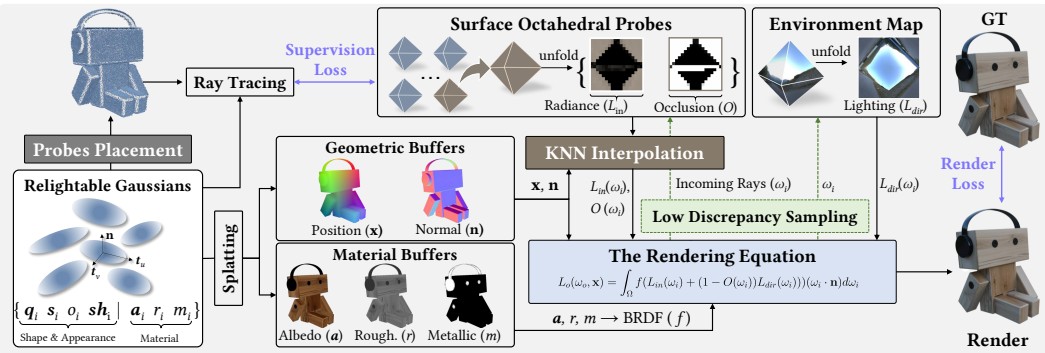

Figure 3: **Inverse Rendering with Surface Octahedral Probes (SOPs).** We utilize trained relightable 2D Gaussians to generate GBuffers via splatting, followed by deferred physically based rendering for a render image. Illumination is split into direct lighting from environment map, indirect lighting and occlusion captured by textures in SOPs. Both the environment map and textures are stored as octahedral textures. Low-discrepancy ray sampling with random rotation is used to compute illumination at shading point, with indirect light and occlusion derived via KNN interpolation from nearby probes. SOPs are initialized with ray tracing and optimized under its guidance, avoiding intensive ray tracing per optimization iteration and boosting inverse rendering efficiency.

To enable relightable 2D Gaussians, we further equip them with material parameters $\{\mathbf{a}_i, r_i, m_i\}$ for albedo, roughness, and metallic. Through 2DGS, we perform differentiable multi-target rendering to generate G-buffers $\mathcal{B}$ in a single pass via alpha blending:

$$\mathcal{B} = \sum_{i=1}^{N} T_i \alpha_i b_i, \quad T_i = \prod_{j}^{i-1}(1 - \alpha_j), \tag{1}$$

where $b_i = \{\mathbf{c}_i, 1, d_i, \mathbf{n}_i, \mathbf{a}_i, r_i, m_i\}$, with $\mathbf{c}_i$ as color, $d_i$ as ray-surfel intersection depth and $\mathbf{n}_i$ as surfel normal. These G-buffers includes RGB buffer $\mathcal{C}$, geometry buffers ($\mathcal{W}$ for weight, $\mathcal{D}$ for depth, $\mathcal{N}$ for normal) and material buffers ($\mathcal{A}$ for albedo, $\mathcal{R}$ for roughness, and $\mathcal{M}$ for metallic). As in prior works (Gao et al., 2024; Liang et al., 2024b), we use the unbiased depth $\tilde{\mathcal{D}} = \mathcal{D}/\mathcal{W}$.

### 3.1.1 STEP 1: RADIANCE AND GEOMETRY RECONSTRUCTION

**Loss** We follow the rendering loss from (Kerbl et al., 2023):

$$\mathcal{L}_{rgb} = L_1(\mathcal{C}, \hat{\mathcal{C}}_{gt}) + 0.2 \cdot (1 - SSIM(\mathcal{C}, \hat{\mathcal{C}}_{gt})), \tag{2}$$

and apply depth-normal consistent regularization (Huang et al., 2024; Gao et al., 2024):

$$\mathcal{L}_{d2n} = 1 - (\mathcal{N} \cdot \mathcal{N}_d), \tag{3}$$

where $\mathcal{N}_d$ is devided from depth. For objects, we apply mask constraint (Gao et al., 2024) as:

$$\mathcal{L}_{mask} = -\mathcal{K} \log \mathcal{W} - (1 - \mathcal{K}) \log (1 - \mathcal{W}), \tag{4}$$

where $\mathcal{K}$ is provided mask, The total loss for the first step is:

$$\mathcal{L} = \mathcal{L}_{rgb} + \lambda_{d2n}\mathcal{L}_{d2n} + \lambda_{mask}\mathcal{L}_{mask}. \tag{5}$$

### 3.1.2 STEP 2: MATERIAL AND LIGHTING DECOMPOSITION

**Deferred Physically Based Rendering** From G-Buffers, we produce a PBR image in a deferred shading manner. We first project the unbiased depth map $\tilde{\mathcal{D}}$ into world space as shading points $\{\mathbf{x}\}$, and then evaluate the rendering equation at these shading points:

$$L_o(\omega_o, \mathbf{x}) = \int_{\Omega} f(\omega_o, \omega_i, \mathbf{x}) L_i(\omega_i)(\omega_i \cdot \mathbf{n}) d\omega_i, \tag{6}$$

where $\mathbf{n}$ is the surface normal, $f$ denotes the BRDF, $L_i$ and $L_o$ are the incoming and outgoing radiance from direction $\omega_{\mathbf{i}}$ and $\omega_{\mathbf{o}}$, and $\Omega$ represents the hemisphere above the surface. We adopt a simplified Disney BRDF model (Burley & Studios, 2012). We evaluate this integral via Monte Carlo sampling by drawing $S_r$ rays $\{\omega_i\}_{i=1}^{S_r}$ from a low-discrepancy Hammersley point set, computed as:

$$\mathcal{C}_{pbr}(\mathbf{x}) = \frac{2\pi}{S_r} \sum_i^{S_r} f(\omega_o, \omega_i, \mathbf{x}) L_i(\omega_i)(\omega_i \cdot \mathbf{n}). \tag{7}$$

**Illumination Modeling**  We model illumination as direct lighting $L_{dir}$ and indirect lighting $L_{in}$, combined through occlusion $O$:

$$L_i(\omega_i) = (1 - O(\omega_i))L_{dir}(\omega_i) + L_{in}(\omega_i). \tag{8}$$

We set direct lighting as a learnable environment map, and the indirect lighting as the inter-reflection. A straight-forward method to obtain occlusion and indirect lighting is to perform ray tracing on Gaussian point cloud, similar to IRGS (Gu et al., 2025). However, ray tracing is computationally intensive, resulting in low efficiency.

To address this, we propose **Surface Octahedral Probes (SOPs)** for efficient querying of indirect lighting and occlusion. These probes are positioned near the surface, with their radiance and occlusion textures $\{L_{in}, O\}$ initialized via ray tracing. In implementation, we employ octahedral textures (Praun & Hoppe, 2003) for their low memory footprint and minimal distortion.

**Automatic Placement of SOPs**  Careless placement can mix SOPs with Gaussian points, causing light-leaking artifacts. To mitigate this issue, we propose a placement strategy that positions SOPs near the surface. We first render geometry buffers for all viewpoints, followed by Multi-View Depth Fusion (Galliani et al., 2015) to generate a dense surface point cloud with normals. We then apply Farthest Point Sampling (FPS) to obtain a uniform subsample of points. Finally, these points are slightly offset along their normals to define the placement locations of SOPs.

**Efficient Querying through SOPs**  At each shading point $\mathbf{x}$, we efficiently query indirect lighting and occlusion from SOPs via K-Nearest Neighbors (KNN) interpolation. As an example, the indirect lighting at $\mathbf{x}$ is queried by first identifying neighboring SOPs $k \in N(\mathbf{x})$ using Fixed Radius Nearest Neighbors (FRNN) search, followed by interpolation:

$$L_{in}(\mathbf{x}) = \frac{\sum_k w_s(k) w_b(k) \cdot L_{in}(k)}{\sum_k w_s(k) w_b(k)}, \tag{9}$$

where $w_s$ and $w_b$ is the spatial and back-face weights, respectively. Given $\mathbf{p}_k$ as the location of $k$-th neighbor SOP, we define a direction vector as $\mathbf{d}_k = \mathbf{p}_k - \mathbf{x}$. Then, the spatial weight is defined as:

$$w_s = \frac{1}{\|\mathbf{d}_k\|}, \tag{10}$$

which assign greater influence for SOPs closer to the shading point. And, the back-face weight (Majercik et al., 2019; McGuire et al., 2017) is defined as:

$$w_b = 0.5 \cdot (1 + \frac{\mathbf{d}_k}{\|\mathbf{d}_k\|} \cdot \mathbf{n}_p) + 0.01, \tag{11}$$

indicating greater contribution when the direction vector $\mathbf{d}_k$ is more aligned with the normal $\mathbf{n}_p$.

**Loss**  We use a rendering loss similar to that in Eq. 2 for the PBR image $\mathcal{C}_{pbr}$, denoted as $\mathcal{L}_{pbr}$. For material regularization, we adopt the Lambertian assumption (Yao et al., 2022) with a cost favoring high roughness and low metallic values under non-view-dependent lighting:

$$\mathcal{L}_{lam} = L_1(\mathcal{R}, 1) + L_1(\mathcal{M}, 0). \tag{12}$$

We also supervise SOPs' textures $\{L_{in}, O\}$ using the traced radiance $L_{tr}$ and occlusion $O_{tr}$:

$$\mathcal{L}_{sops} = L_1(L_{in}, L_{tr}) + L_1(O, O_{tr}). \tag{13}$$

The final loss for the second step is:

$$\mathcal{L} = \mathcal{L}_{pbr} + \lambda_{lam}\mathcal{L}_{lam} + \lambda_{sops}\mathcal{L}_{sops} + \lambda_{d2n}\mathcal{L}_{d2n} + \lambda_{mask}\mathcal{L}_{mask}, \tag{14}$$

## 3.2 EDITING

From reconstructed scene and relightable Gaussian object, the editing stage estimates scene lighting and caches occlusion, bridging to relighting and real-time shadow computation during rendering.

### 3.2.1 LIGHTING ESTIMATION

We assume that the object is small relative to the scene and that its placement affects only its own appearance and nearby regions. Under this assumption, the challenging task of estimating lighting in complex scenes can be reformulated as a more tractable environment map inpainting problem.

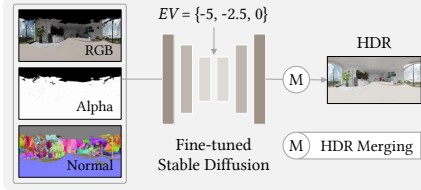

From a reconstructed Gaussian scene, we generate an HDR environment map for the target placement location (Figure 4). We first perform a 360° sweep around the location to capture an incomplete RGB panorama, a partial normal map, and an alpha mask of the reconstructed regions. These inputs are processed by a fine-tuned Stable Diffusion 2.1 (Rombach et al., 2022) model to generate a complete environment map. For HDR, we use an exposure value (EV)-conditioned prompting scheme (Phongthawee et al., 2024): a base model is trained at $EV = 0$ and fine-tuned with interpolated embeddings for different exposure levels. During inference, we generate maps at $EV = \{-5, -2.5, 0\}$ and fuse them into an HDR environment map, which is then converted to an octahedral texture for relighting and shadow computation. More details are included in Appendix D.

Figure 4: **Lighting Estimation.** At a given location, we create a partial panoramic view via a 360° sweep of the Gaussian scene, yielding an incomplete RGB image, normal map, and alpha mask of reconstructed areas. Then, we use a fine-tuned Stable Diffusion to infer a HDR environment map.

### 3.2.2 OCCLUSION CACHING AND SHADOW CASTING

The shadow cast by a newly placed object in the scene is determined by the occlusion $O'$ it introduces. Computing $O'$ directly via ray tracing for every rendering is straightforward but computationally expensive. Instead, we cache the object-induced occlusion $O'$ using our proposed SOPs.

Under the Lambertian scene assumption, the rendering equation (Eq. 6) simplifies to:

$$L_o \approx f_d \int L_i(\omega_i)(\omega_i \cdot \mathbf{n}) d_{\omega_i}. \tag{15}$$

When an object is placed, the scene rendering becomes:

$$L'_o = f_d \int L_i(\omega_i)(1 - O'(\omega_i))(\omega_i \cdot \mathbf{n}) d_{\omega_i}, \tag{16}$$

where $O'$ is the object-induced occlusion cached by SOPs newly placed in the scene, distinct from the self-occlusion $O$ of the object in Eq. 8. The cast shadow is then derived as:

$$\mathcal{S} = \frac{L'_o}{L_o}. \tag{17}$$

To implement this, a potential shadow region is defined around the object placement location, with size proportional to $N$ times the object dimensions. Within this region, SOPs are distributed following the *Automatic Placement* strategy (Sec. 3.1.2). Their occlusion textures are then precomputed via ray tracing, enabling efficient shadow calculation during rendering.

## 3.3 RENDERING

We start by performing multi-target rendering (Sec. 3.1) to obtain the RGBD of the Gaussian scene. Next, we relight the object using the estimated environment map (Sec. 3.2.1) and compute shadows with cached occlusion (Sec. 3.2.2).

Both object relighting and shadow computation involve evaluating integrals, which we accelerate via *Monte Carlo importance sampling*. The octahedral texture's low distortion and roughly uniform texel areas simplify defining the Probability Density Function (PDF) as:

Table 1: **Composition Performance on SynCom Dataset.** Objective metrics (PSNR, SSIM), subjective metrics (3D consistency, Con.; harmony, Harm.), and efficiency metrics (editing time, FPS) are reported.

| | PSNR↑ | SSIM↑ | Con.↑ | Harm.↑ | FPS↑ | T(Edit)↓ |
|---|---|---|---|---|---|---|
| DiffHarmony | 22.436 | 0.825 | 3.125 | 2.929 | 0.01 | - |
| ZeroComp | 20.344 | 0.780 | 1.642 | 1.504 | 0.40 | - |
| MV-CoLight | 21.045 | 0.855 | 2.800 | 2.633 | 1.01 | - |
| GS-IR | 22.418 | 0.824 | 3.283 | 2.125 | 2.11 | - |
| GI-GS | 22.877 | 0.849 | 3.746 | 2.908 | 0.29 | - |
| IRGS | 22.417 | 0.799 | 3.496 | 2.883 | 0.03 | - |
| DiffusionLight | 21.842 | 0.841 | 1.913 | 2.171 | 0.02 | - |
| Ours (Trace) | **24.567** | **0.870** | **4.746** | **4.600** | 4.02 | 14.59 |
| Ours (SOPs) | 24.282 | 0.868 | 4.563 | 4.588 | **26.14** | 36.12 |

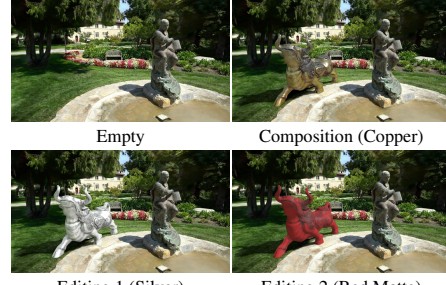

Empty — Composition (Copper)

Editing 1 (Silver) — Editing 2 (Red Matte)

Figure 5: **Material Editing** in real-world composition, from *copper* to *silver* and *red matte*. Please ZOOM IN for details.

$$PDF(\omega_i) = \frac{1}{4\pi} H_t \cdot W_t \cdot I. \qquad (18)$$

where $H_t$, $W_t$, and $I$ denote the height, width, and intensity of the texture, respectively.

Our relighting process begins by sampling a set of rays $S_r$ from the probability density function (PDF) in Eq. 18, followed by Monte Carlo integration of the rendering equation (Eq. 6). To account for self-occlusion, we modulate the direct lighting $L_{dir}(\omega_i)$ by the visibility factor $(1 - O(\omega_i))$ derived from the object's SOPs, yielding $L_i(\omega_i) = (1 - O(\omega_i))L_{dir}(\omega_i)$. This adjusted term is then used in the integration (Eq 7). As for indirect lighting $L_{in}$, we intentionally omit it due to its prohibitive computational cost. Then, cast shadows are subsequently computed via Eq. 17, where the occlusion $O'$ is interpolated from the SOPs placed in the scene using *Efficient Querying* (Sec. 3.1.2). Finally, depth compositing is applied to produce a realistic 3D object-scene composition.

# 4 EXPERIMENTS

## 4.1 IMPLEMENTATION DETAILS

**Reconstruction Details** We set the octahedral environment map resolution to $256 \times 256$, and SOPs' texture resolutions to $16 \times 16$, using 128 sampling rays. SOPs number is set to 5k, with an offset distance empirically set to 1% of the object's size. At step 1, loss weights are $\lambda_{d2n} = \lambda_{mask} = 0.05$, with other settings following 2DGS, training over 30k iterations. At step 2, learning rates are 0.01 for environment map, albedo, roughness, and metallic, and 0.001 for SOPs textures, initialized via 2D ray tracing; loss weights are $\{\lambda_{lam}, \lambda_{sops}\} = \{0.001, 1\}$, over 2k iterations.

**Editing and Rendering Details** The estimated environment map resolution is $1024 \times 512$, converted to an octahedral texture at $512 \times 512$. We define the scene within 6 times the object's size as the potential shadow region, placing 10k SOPs there. We sample 256 rays for rendering.

## 4.2 COMPOSITION PERFORMANCE

**SynCom Dataset** We create a synthetic dataset to evaluate 3D object-scene composition. Multi-view images of 4 objects (*bottle*, *horse*, *kettle*, and *toy*), 4 scenes (*artwall*, *attic*, *forest*, *room*), and their 16 compositions are rendered using the Blender Cycles Engine. The precise control of object placement and camera settings in Blender ensures a highly reliable reference for evaluation. Further details of our SynCom dataset are provided in Appendix A.

For the composition performance evaluation, we compare three distinct categories of approaches:

(1) **Image Composition**: Scene and object are first reconstructed using 2DGS (Huang et al., 2024), synthesized for novel views, and then composed with a image composition algorithm, specifically DiffHarmony (Zhou et al., 2024) and ZeroComp (Zhang et al., 2025). Although MV-CoLight (Ren et al., 2025) is a two-stage method that first performs image composition and then addresses 3D composition, we categorize it here for simplicity.

Table 2: **Reconstruction performance on TensoIR.** Our method achieves accuracy comparable to SOTA approaches with at least **2×** efficiency improvement.

| Method | NVS PSNR↑ | Albedo PSNR↑ | Relighting PSNR↑ | Training Time↓ |
|--------|-----------|--------------|------------------|----------------|
| NeRFactor | 24.679 | 25.125 | 23.383 | >100 h |
| InvRender | 27.367 | 27.341 | 23.973 | 15 h |
| TensoIR | 35.088 | 29.275 | 28.580 | 5 h |
| GS-IR | 35.333 | 30.286 | 24.374 | 16.40 min |
| R3DG | **38.423** | **31.926** | 29.766 | 47.80 min |
| IRGS | 35.751 | 31.658 | 30.250 | 21.45 min |
| Ours | 35.822 | 31.683 | **30.474** | **7.93 min** |

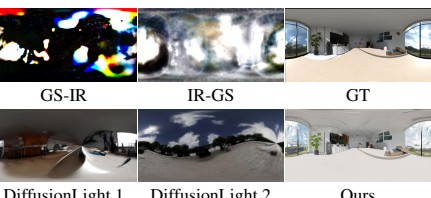

Figure 6: **Environment Maps Comparison.** GS-IR and IRGS fail in complex scenes, DiffusionLight is viewpoint-inconsistent, while our method yields superior and consistent results.

(2) **Gaussian-based Inverse Rendering**: Scene and object are reconstructed with Gaussian-based inverse rendering methods, composited, and then rendered with PBR. Specifically, GS-IR (Liang et al., 2024b), GI-GS (Chen et al., 2025) and IRGS (Gu et al., 2025) are selected for comparison. Note that R3DG (Gao et al., 2024) is not included due to excessive memory usage on scenes, resulting in reconstruction failure.

(3) **Variants of Our Pipeline**: (a) *DiffusionLight*, which estimates illumination from a single RGB image via the DiffusionLight (Phongthawee et al., 2024); (b) *Our (Trace)*, which employs computationally intensive ray tracing on Gaussians to generate shadows; (c) *Our (SOPs)*, which utilizes our proposed SOPs to cache occlusion and achieve efficient shadow rendering.

We report both objective and subjective metrics in Table 1. For objective evaluation, Peak Signal-to-Noise Ratio (PSNR) and Structural Similarity Index (SSIM) are computed against the ground truth. Subjective evaluation is based on a Mean Opinion Score (MOS) survey, where 40 participants rate 3D consistency (Con.) and harmony (Harm.) of the compositions from 1 (poor) to 5 (excellent). Editing time and rendering frame rate are also reported.

We show the composition results of all methods in Figure 7. Across both objective and subjective metrics, as well as overall rendering performance, our methods *Our (Trace)* and *Our (SOPs)* consistently outperform competing approaches, delivering harmonious compositions and, importantly, realistic shadows. In particular, *Our (Trace)* yields smoother shadows but at a lower rendering speed, whereas *Our (SOPs)* achieves comparable quality while sustaining a significantly higher frame rate.

**Real-World Datasets** Beyond synthetic data, we further validate our pipeline on real-world object–scene composition. For **public datasets**, we select four objects from BlendedMVS (Yao et al., 2020) and four scenes from Tanks and Temples (Knapitsch et al., 2017), with the results shown in Fig. 11. We also evaluate on **smartphone-captured sequences**, including two objects and two scenes reconstructed using COLMAP (Schonberger & Frahm, 2016), as illustrated in Fig. 14. Overall, these results demonstrate that our approach generalizes well to real-world data, producing harmonious compositions with realistic shadows.

### 4.3 RECONSTRUCTION PERFORMANCE

Although our main focus is realistic 3D object–scene composition, we also assess reconstruction accuracy on TensoIR (Jin et al., 2023) and our SynCom-Object dataset. As shown in Table 2, our approach achieves accuracy comparable to state-of-the-art methods while delivering the fastest reconstruction, thanks to the proposed SOPs. R3DG attains the highest novel view synthesis accuracy due to its lack of indirect lighting supervision, but this causes *incorrect indirect lighting* and *bright spot artifacts* in relighting, as shown in Figure 19. Compared with IRGS, our method avoids expensive per-point ray tracing through efficient KNN interpolation and trains on the full image instead of random pixel sampling. Quantitative results on SynCom-Object (Table 4) further validate the accuracy and efficiency of our method. Additional results are provided in Appendix C.

### 4.4 LIGHTING ESTIMATION PERFORMANCE

We present environment maps estimated by different methods in Figure 6. Inverse rendering approaches, such as GS-IR and IRGS, struggle to capture the intricate lighting transports in com-

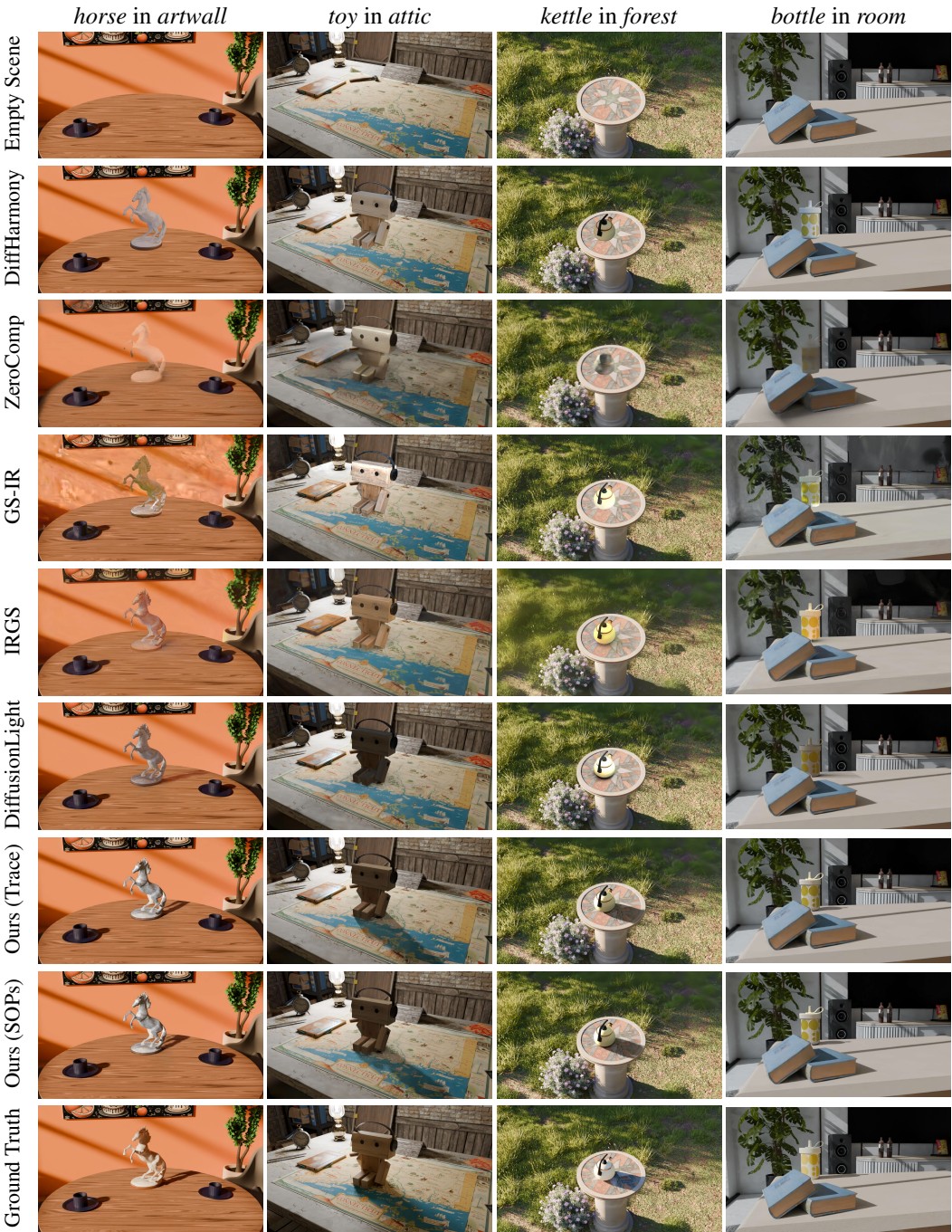

Figure 7: **Object-scene composition results on SynCom dataset.** Our methods, *Ours (Trace)* and *Ours (SOPs)*, outperform others in realistic object-scene composition, producing harmonious visuals and realistic shadows. *Ours (Trace)* offers slightly better quality but at a lower rendering frame rate. DiffHarmony and ZeroComp, based on image-level harmonization or composition, struggles with scene-object occlusion. Inverse rendering methods like GS-IR and IRGS deviate from real lighting in complex scenes: GS-IR's coarse model lacks harmony, while IRGS, though more detailed, fails to render realistic shadows. DiffusionLight generates shadows in some cases but suffers from instability due to inconsistent single-image estimations across different viewpoints.

plex scenes. This difficulty is further exacerbated by the incomplete coverage of captured views. Learning-based methods such as DiffusionLight, rely on a single input image and often produce inconsistent lighting estimations across different viewpoints. By contrast, our method leverages the reconstructed scene radiance field, enabling both higher-quality lighting estimation and multi-view consistent object–scene composition. Further results are presented in Appendix D.

## 5 DISCUSSION

**Assumptions and Failure Cases**  3D object-scene composition is a highly challenging problem, and we rely on several assumptions (Ye et al., 2024b; Wang et al., 2022b) to make it tractable.

- The inserted object is relatively small and affects only the local area of the scene.
- The scene is predominantly Lambertian, allowing a reasonable approximation of shadows.

These assumptions naturally limit the applicability of our approach. As illustrated in Figure 8, two representative failure cases highlight these limitations.

In the first case, the object fails to cast a remote shadow, which is due to the first assumption. In the second case, we modify the table in the *room* scene from our SynCom dataset to be highly specular, which prevents our method from accurately modeling mirror-like reflections. This failure stems from both the second assumption and the limitations of the 2DGS-based reconstruction stage (Section 3.1.1), which struggles with reflective scenes.

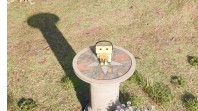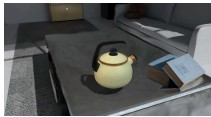

(a) Remote shadow  (b) Mirror reflection

Figure 8: **Failure cases.** We failed to cast remote shadow and model mirror-like reflections.

**SOPs Placement**  In our experiments, we place SOPs using a heuristic approach to reduce light leaking, by slightly offsetting each SOP along the surface normal. In all experiments, this offset is set to 1% of the object size. This is validated to some extent in Figure 20 and Table 5, and we did not observe any noticeable failures or severe light leaks in all our experiments. We also find that the shadow quality improve as the number of SOPs and the resolution of their textures increase in Figure 15. Building on these insights, future work could explore adaptive SOP placement strategies to achieve better fidelity with fewer SOPs, and reduce memory usage.

**Incremental SOP Updates**  Our pipeline naturally handles camera motion, since SOPs cache occlusion in scene space and can be reused across views. When the inserted object moves, changes in visibility and lighting require recomputing SOPs. Developing efficient incremental update strategies for moving objects remains challenging and is an interesting direction for future work.

**Single-View Scenes**  Our method relies on multi-view 3D reconstruction, but extending it to single-image inputs is an interesting direction for future work.

## 6 CONCLUSIONS

In this study, we tackle the challenge of **realistic 3D object-scene composition**, seamlessly integrating objects into scenes with *visual harmony* and *physically plausible shadowing*. We propose ComGS, organized into three stages: reconstruction, edit, and rendering. In the *reconstruction* stage, we reconstruct relightable Gaussian objects and the scene's Gaussian radiance field. Surface Octahedral Probes (SOPs) are introduced to accelerate object reconstruction without compromising accuracy. In the *edit* stage, within the context of object-scene composition, we simplify complex scene lighting estimation to a local lighting completion problem at the object placement site, solved via a fine-tuned diffusion model. SOPs are also used to cache occlusion caused by the inserted objects. In the *rendering* stage, we perform object relighting and shadow computation, combining results through depth compositing to produce the final object-scene composition. Our framework enables near-real-time rendering with vivid and physically plausible shadows.

**Acknowledgments**  This work was supported by National Key R&D Program of China (2023YFB3209702), the National Natural Science Foundation of China (62472213), and was partially supported by Jiangsu Broadcasting Corporation.

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

# A  DETAILS OF SYNCOM DATASET

## A.1  OVERVIEW

The task of **realistic 3D object–scene composition** aims to reconstruct objects and scenes separately from two sets of multi-view images, and then integrate the reconstructed 3D objects into the 3D scenes in a visually harmonious manner. *Realism* here means that the composed objects match the scene in appearance and produce physically plausible shadows.

To enable quantitative evaluation, we introduce **SynCom** (Synthetic Composition), a synthetic dataset designed for this task. Compared to real-world data, synthetic data offers precise control over object placement and camera setting, ensures accurate ground-truth for the composed scenes, and provides high reproducibility, all of which facilitate reliable quantitative assessment. Below, we provide a detailed description of the SynCom dataset.

## A.2  DATA SOURCE

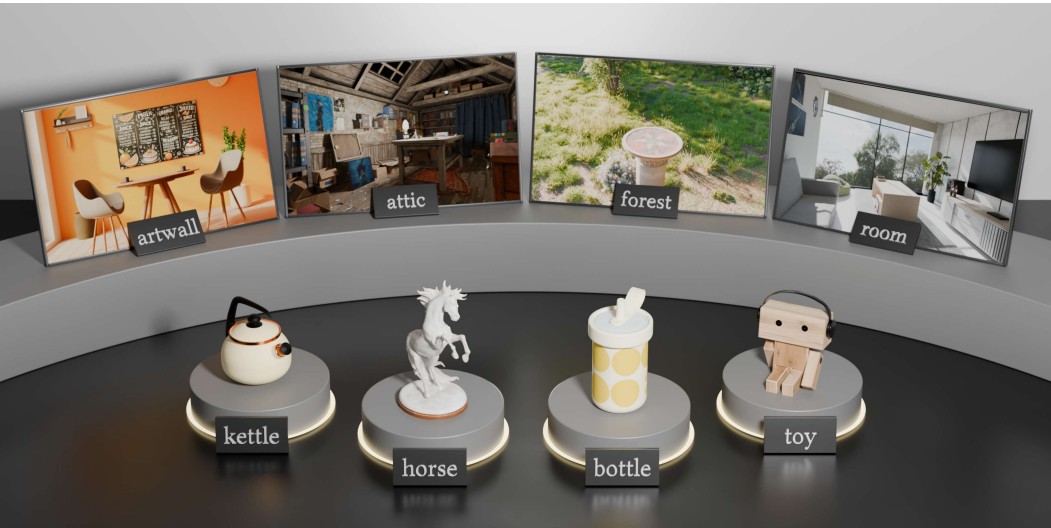

Figure 9: Assets collected from BlenderKit, serving as the data source for our SynCom dataset, comprising four objects (*bottle*, *horse*, *kettle*, *toy*) and four scenes (*artwall*, *attic*, *forest*, *room*).

We collect 4 object assets and 4 scene assets from BlenderKit[1], all under permissive licenses. The assets are shown in Figure 9. The object assets are designated as *bottle*, *horse*, *kettle*, and *toy*, while the scenes are labeled as *artwall*, *attic*, *forest*, and *room*. To facilitate camera placement and approximate realistic environments, we perform careful manual editing of the scenes, adjusting layouts and enriching content.

## A.3  SYNCOM DATASET

Our SynCom dataset is organized into three distinct collections: *object*, *scene* and *composition*. We render the dataset using the Cycles engine in Blender[2] to achieve high physical realism, particularly accurate global illumination and shadow effects.

**Object Collection**   This collection is created by rendering each of the four objects individually. The objects are illuminated using freely available 1K-resolution HDRIs from PolyHaven[3] as environment maps. The collection is divided into a training set and a testing set. The training set focuses on the upper hemisphere of each object, with 200 camera positions randomly sampled around this region. The testing set comprises 100 images captured from three distinct latitude circles around

---

[1]https://www.blenderkit.com/

[2]https://www.blender.org/

[3]https://polyhaven.com/

the object. For all images, cameras point toward the object center, and the resolution is fixed at $800 \times 800$ pixels.

**Scene Collection**  This collection focuses on rendering each of the four scenes individually. To create high-fidelity and complex scenes, we start with 4 scenes sourced from BlenderKit and modify them by adding, removing, or altering elements. This process results in 4 distinct environments with real-world-like complexity: 3 indoor settings (*artwall*, *attic*, *room*) and 1 outdoor setting (*forest*).

The process of rendering scenes is inherently more complex than rendering individual objects. Capturing a scene requires moving the camera inside it, which can lead to occlusion by the scene's structures or even cause the camera to intersect with them, resulting in invalid views. To prevent this, We employ a virtual auxiliary ellipsoid, with scales and center carefully adjusted along each axis, so that its upper surface remains clear of the scene's structures. Cameras are then placed on the ellipsoid's surface to capture the scene. Compared to a spherical boundary, an ellipsoid provides greater flexibility in accommodating the diverse shapes and layouts of different environments.

For the training set, 144 camera positions are randomly sampled from cells of a uniform grid on the ellipsoid's surface, with all cameras pointing toward its center. For the test set, 72 camera positions are sampled along a spiral path on a slightly smaller, concentric ellipsoid. All images are captured at a resolution of $1280 \times 720$ pixels.

**Composition Collection**  This collection combines the four objects with the four scenes, yielding 16 distinct object–scene pairs. Each object is manually positioned with specified 3D *location*, *orientation*, and *scale*, and these placement parameters are recorded for later use. The ground truth for each composition is obtained by rendering the scene after object placement. To ensure controlled evaluation, all composite scenes are rendered from the same 72 test viewpoints as their corresponding empty scenes.

Leveraging the controllability of synthetic data, these renderings provide a reliable benchmark for assessing realistic object–scene composition and related tasks such as object insertion, novel-view synthesis, and inverse rendering.

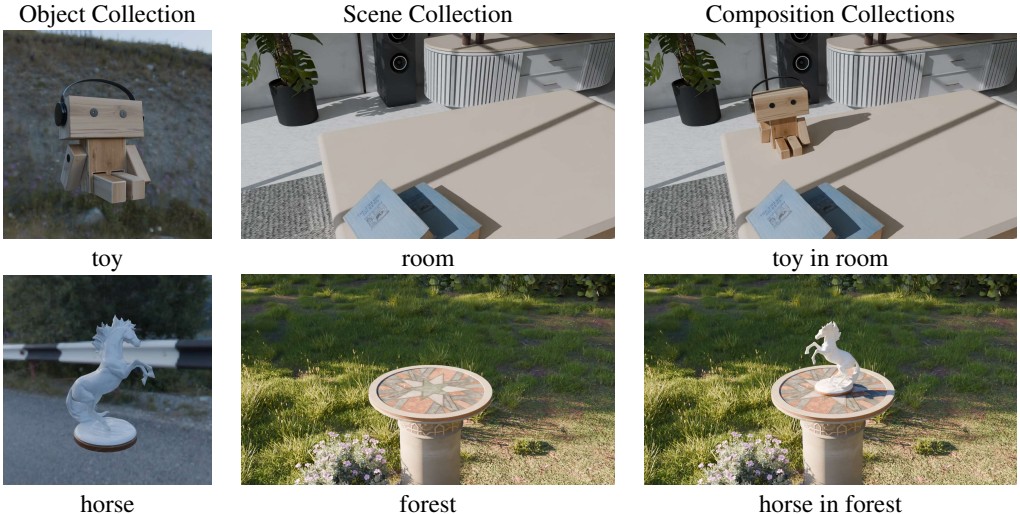

Figure 10: **Samples selected from the SynCom dataset.** The three columns from left to right correspond to the Object, Scene, and Composition collections. The precise control over cameras and the highly controllable scene illumination in synthetic data enable alignment between empty and composed scenes, facilitating quantitative evaluation of realistic 3D object–scene composition.

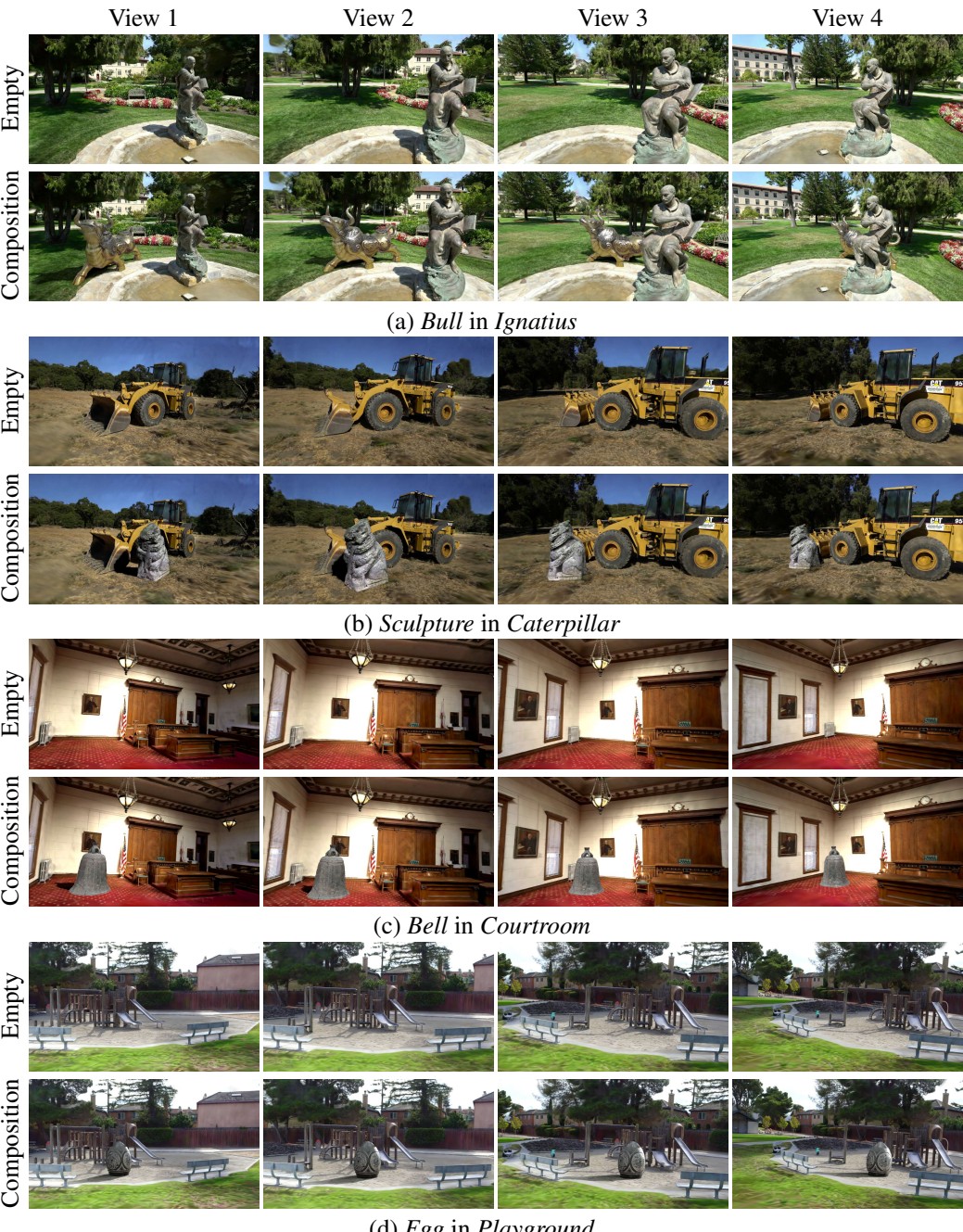

Figure 11: **3D object-scene composition on public datasets**. We select four scenes (*Ignatius*, *Caterpillar*, *Courtroom* and *Playground*) from the Tanks and Temples dataset and four objects (*Bull*, *Sculpture*, *Bell* and *Egg*) from the BlendedMVS dataset. Starting from the multi-view images provided by these datasets, we perform 3D object–scene composition using our pipeline, achieving harmonious composition of objects into the scenes along with physically plausible cast shadows.

# B MORE RESULTS ON COMPOSITION

## B.1 PUBLIC DATASETS

In addition to our synthetic SynCom dataset, we also evaluate our method on public datasets. Specifically, we select several scenes from the Tanks and Temples (Knapitsch et al., 2017) and several objects from the BlendedMVS (Yao et al., 2020), with object masks obtained using SAM2 (Ravi et al., 2024). Our method is then applied to perform 3D object–scene composition on these data, as shown in Figure 11. For more intuitive visualizations, please see the supplementary video.

**Material Editing** We further demonstrate *multi-view consistent* material editing of objects in Figure 12. For a clearer visualization of these results, we refer readers to our supplementary video.

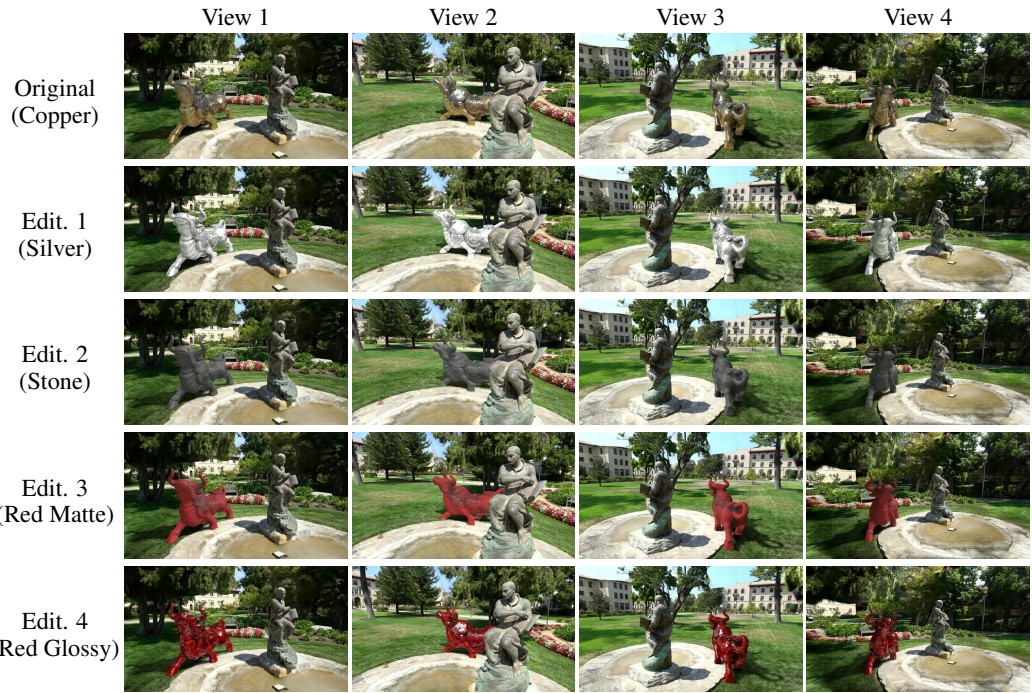

Figure 12: **Material Editing**. Our reconstructed objects are represented as relightable Gaussian point clouds, enabling flexible material editing. By adjusting the *albedo*, *metallic*, and *roughness* properties, we demonstrate material modifications of object from original *copper* to *silver*, *stone*, *red matte*, and *red glossy*. The edited results remain consistent across multiple viewpoints.

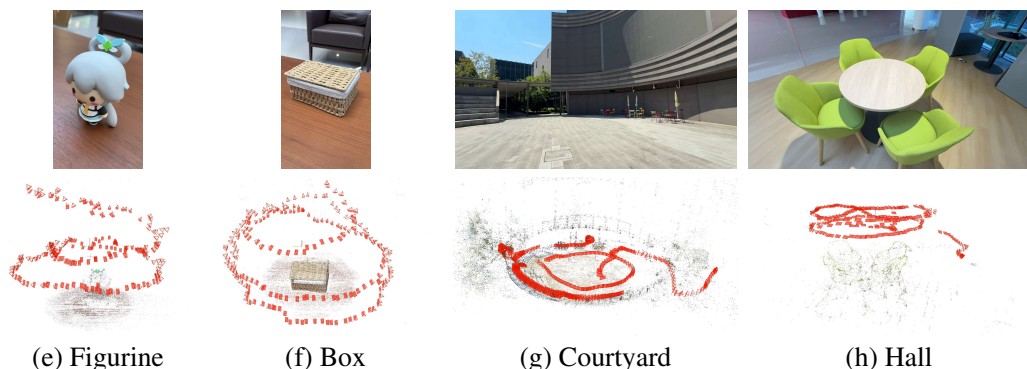

Figure 13: **Smartphone-Captured Objects and Scenes.** Top row: sample images of our captured objects and scenes. Bottom row: corresponding camera pose visualizations.

## B.2 SMARTPHONE-CAPTURED DATA

We capture separate videos of scenes (i.e., an outdoor *courtyard* and an indoor *hall*) and objects (i.e., a *figurine* and a *box*) using an iPhone 16 Pro. Objects were placed under soft lighting and captured following a systematic spiral camera trajectory, which ensures sufficient viewpoint coverage for reconstruction. Figure 13 shows sample images along with visualizations of the captured camera poses, providing an overview of our real-world data collection setup.

Multi-view images of each are independently extracted from their respective videos via frame sampling, while object masks are obtained with SAM2 (Ravi et al., 2024). Our method is then applied to perform 3D object–scene composition. As shown in Figure 14, even on challenging smartphone-captured data, our approach produces harmonious appearances and realistic shadows. For a clearer and more vivid demonstration, please refer to the supplementary video.

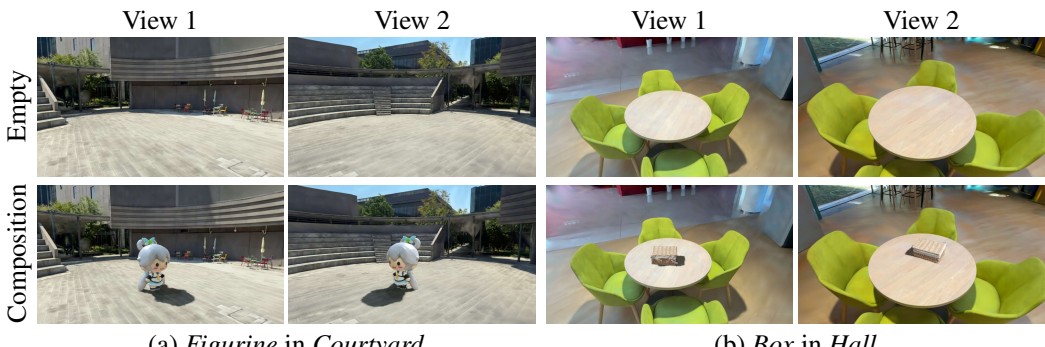

(a) *Figurine* in *Courtyard*      (b) *Box* in *Hall*

Figure 14: **Qualitative results on smartphone-captured data.** Our approach enables 3D object–scene composition with harmonious appearances and realistic shadows, even on challenging smartphone-captured data.

## B.3 MEMORY CONSUMPTION OF SOPS

We evaluate the GPU memory consumption of our SOPs on the *Bottom_in_Room* scene from the SynCom dataset. The results for different numbers of SOPs are summarized in Table 3.

The memory usage with 20K SOPs is comparable to the ray-tracing baseline, which requires maintaining a BVH structure. This indicates a moderate overhead introduced by our method. The mild growth in memory consumption as the number of SOPs increases suggests that the memory is dominated by the Gaussian point cloud, which consists of a large number of Gaussians with high-dimensional attributes.

Table 3: GPU memory usage comparison under different numbers of SOPs.

| Setting | Memory (GB) |
|---|---|
| Ray Tracing (Baseline) | 4.4 |
| SOPs Num. = 10,000 | 4.3 |
| SOPs Num. = 20,000 | 4.4 |
| SOPs Num. = 40,000 | 4.5 |

## B.4 ABLATION ON NUMBERS OF SOPS AND TEXTURE RESOLUTION

We analyze the influence of the number of SOPs and the probe texture resolution within our composition pipeline. Our baseline configuration uses 10k SOPs at a resolution of 16, and is compared against setups with fewer (5k) and more (20k) SOPs, as well as lower (8) and higher (32) texture resolutions.

As illustrated in Figure 15, reducing the number of SOPs to 5k or the resolution to 8 introduces visible shadow aliasing and loss of fine details. Specifically, at a resolution of 8, the angular sampling over the full 360 degrees becomes too coarse, resulting in inadequate shadow representation and significant directional errors. A resolution of 16, by contrast, yields visually acceptable results. Increasing either the number of SOPs or the texture resolution beyond the baseline values reduces aliasing and improves the overall shadow quality.

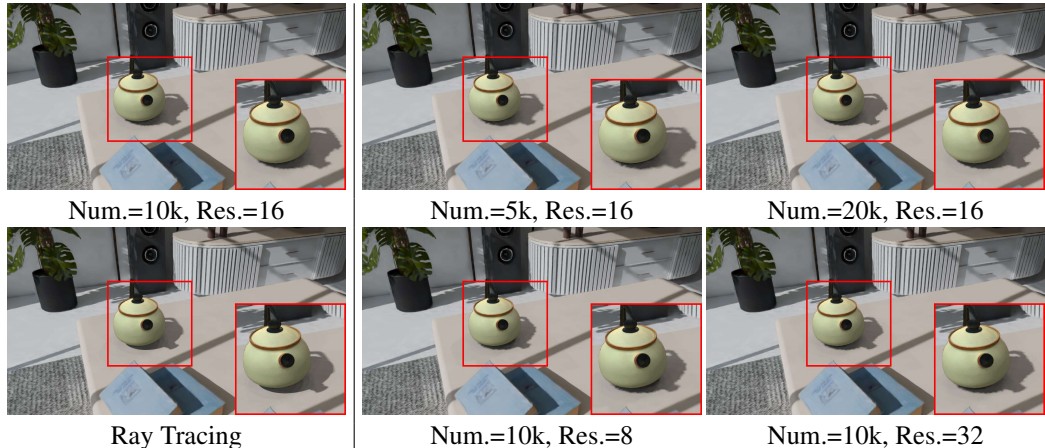

Figure 15: **Influence of the Number of SOPs and Texture Resolution**. A resolution of 16 provides visually acceptable results, while a resolution of 8 causes significant directional errors due to coarse angular sampling. Increasing the number of SOPs beyond 10k further reduces aliasing.

### B.5 SELF-OCCLUSION

Figure 16 presents two composition results for illustrating the effect of modeling self-occlusion. One result is generated without incorporating self-occlusion, and the other is produced by our full method, which models self-occlusion by modulating direct illumination using the visibility factor $1-O(\omega_i)$ computed from the object's SOPs (see Section 3.3). Incorporating self-occlusion provides additional shading cues that contribute to a more realistic 3D object-scene composition.

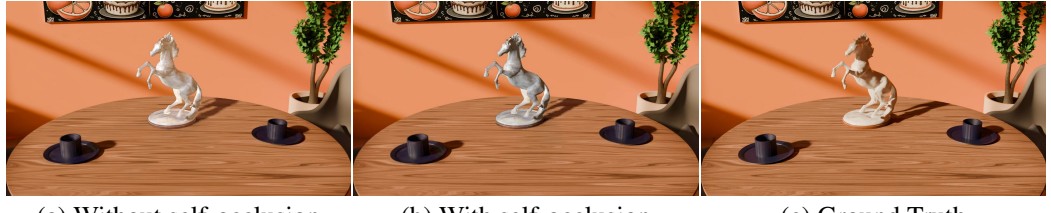

| (a) Without self-occlusion | (b) With self-occlusion | (c) Ground Truth |

Figure 16: **Effect of Self-Occlusion.** Comparison of compositions without and with self-occlusion modeling. Incorporating self-occlusion further enhances the composition's harmony and realism.

### B.6 MULTI-OBJECT COMPOSITION

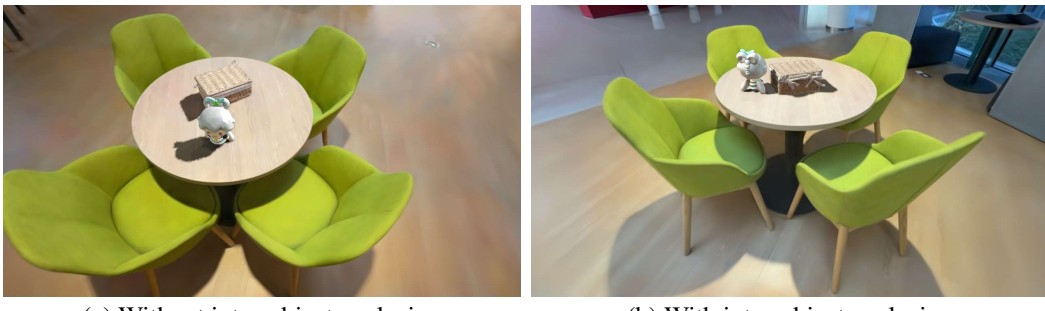

| (a) Without inter-object occlusion | (b) With inter-object occlusion |

Figure 17: **Multi-object Composition in a Scene.** Our approach enables multi-object composition through sequential placement.

Our approach supports multi-object composition through a sequential procedure. Although simultaneous placement of multiple objects is not supported, we address this by inserting objects one after another. After placing the first object, the modified scene is rendered from multiple views and reconstructed using Stage 1 of our method. This reconstructed scene then serves as the base scene for inserting the next object. Figure 17 demonstrates that this sequential strategy allows our method to successfully compose multiple objects and model partial inter-object occlusion between them.

## B.7 SHADOW FROM MULTIPLE LIGHT SOURCES

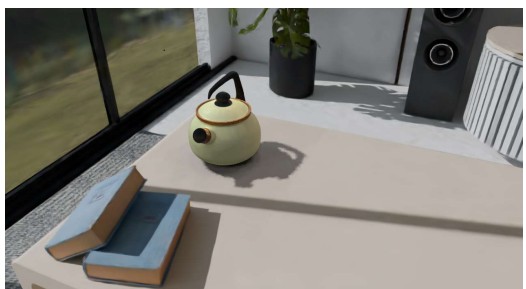 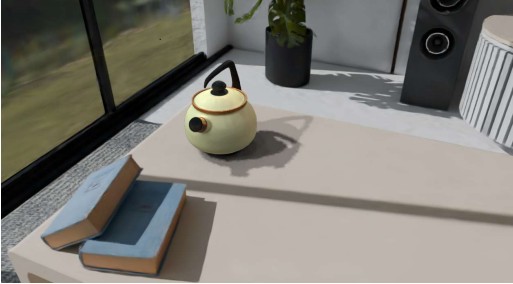

(a) With a light source                 (b) With two light sources

Figure 18: **Shadows from Multiple Lights.** Our method correctly composites shadows in overlapping regions, producing darker, additive shadows when multiple light sources are present.

Our method naturally handles scenes with multiple light sources, producing shadows that correctly composite in overlapping regions. To validate this, we conducted an experiment on the *Kettle_in_Room* scene by manually adding a secondary point light to the predicted environment map.

After introducing the additional light, the shadows cast by the original light become less pronounced due to the increase in overall illumination. In areas where shadows from both light sources overlap, a darker, composite shadow emerges, demonstrating the expected additive effect of multiple light sources. The results are illustrated in Figure 18.

## C MORE RESULTS ON RECONSTRUCTION

### C.1 RECONSTRUCTION PERFORMANCE ON SYNCOM-OBJECT

We also conducted experiments on our SynCom object dataset. We report MAE for normals, PSNR for NVS and albedo, and PSNR, SSIM, and LPIPS for relighting, along with training time. For NVS and albedo, only PSNR is presented due to the negligible differences in SSIM and LPIPS across most methods. Results are presented in Tables 4. Our approach achieves accuracy comparable to SOTA methods, and in some cases slightly surpasses them, while enabling $2\times$ faster reconstruction.

Table 4: Quantitative results on the SynCom-Obj dataset. Our method achieves accuracy comparable to the state of the art while delivering over **2× higher efficiency**.

| Method | Normal | NVS | Albedo | Relighting | | | Training |
|---|---|---|---|---|---|---|---|
| | MAE↓ | PSNR↑ | PSNR↑ | PSNR↑ | SSIM↑ | LPIPS↓ | Time(min)↓ |
| GS-IR (Liang et al., 2024b) | 1.801 | 40.534 | 27.409 | 22.150 | 0.951 | 0.041 | 15.51 |
| R3DG (Gao et al., 2024) | 1.510 | **41.364** | 27.992 | 28.646 | 0.967 | 0.036 | 40.62 |
| IRGS (Gu et al., 2025) | **0.990** | 39.250 | 28.596 | **29.752** | 0.970 | 0.039 | 21.99 |
| Ours | 1.044 | 39.700 | **28.840** | 29.601 | **0.975** | **0.027** | **7.42** |

### C.2 ABLATION ON SOPS INITIALIZATION

To assess the robustness of the heuristic offset used for SOPs initialization, we perform an ablation study by applying different placement offsets (0%, 1%, and 2% of the object size) on the *Toy* scene

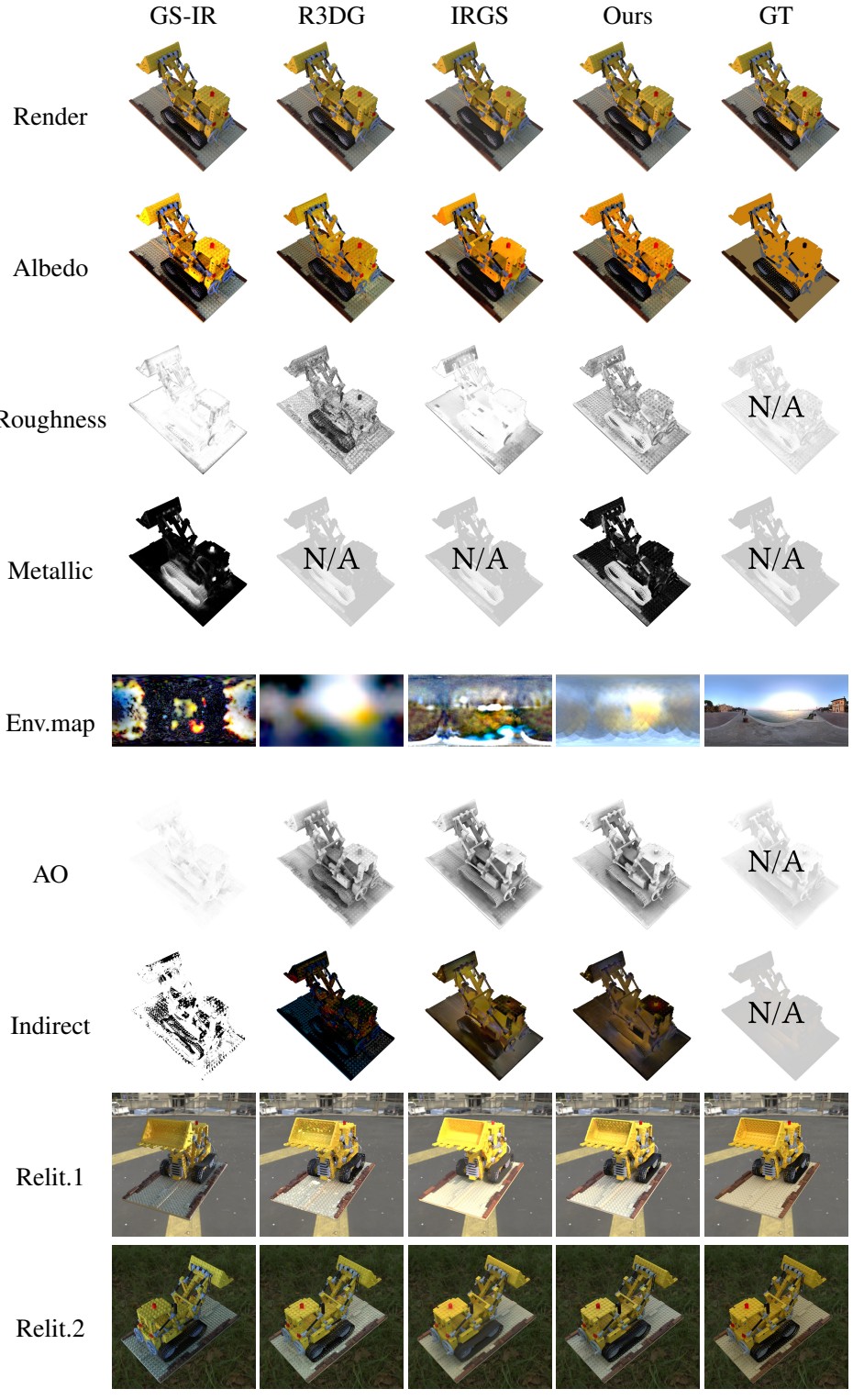

Figure 19: **Qualitative Results on TensoIR Dataset**. Our method achieves effective relightable object reconstruction, enabling satisfactory relighting. In contrast, GS-IR's oversimplified lighting assumptions lead to inaccurate decomposition. R3DG lacks supervision on indirect lighting, leading to *unnatural indirect lighting* and *bright spot artifacts* in relighting.

from the SynCom dataset. Quantitative results are summarized in Table 5. It can be observed that the 1% offset consistently yields the best performance across all metrics, including material albedo estimation, rendering fidelity, and relighting quality.

In addition, we include visual comparisons of relighting and ambient occlusion under different offsets in Figure 20. The visual results demonstrate that the 0% offset leads to noticeable light leakage in both occlusion and relighting, whereas the 1% setting produces clean and stable outcomes. Based on these findings, we adopt the 1% offset strategy uniformly in all experiments reported in this work, and we did not observe any notable failures under this configuration.

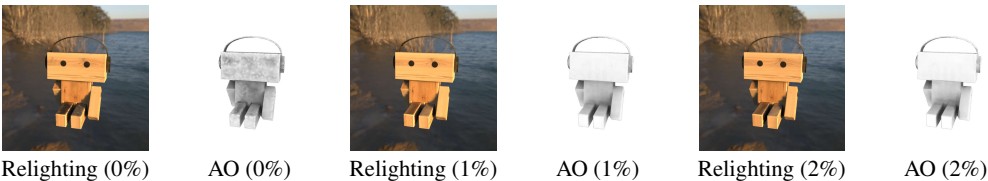

| Relighting (0%) | AO (0%) | Relighting (1%) | AO (1%) | Relighting (2%) | AO (2%) |

Figure 20: **Comparison of relighting and ambient occlusion (AO) under different SOPs initialization offsets.** The 0% offset introduces noticeable light leakage, while the 1% setting produces clean and stable results.

Table 5: **Quantitative ablation study on the SOPs initialization offset.**

| Offset | Albedo PSNR | Rendering PSNR | Relight PSNR |
|--------|-------------|----------------|--------------|
| 0% | 33.222 | 37.204 | 30.449 |
| 1% | **33.304** | **37.265** | **31.324** |
| 2% | 33.130 | 37.202 | 31.281 |

## D  DETAILS OF LIGHTING ESTIMATION

### D.1  QUESTION STATEMENT

Since the scene's geometry and radiance field have already been reconstructed, and our task only requires local illumination around the inserted object for relighting and shadow casting, the inherently complex problem of lighting estimation in the full scene can be significantly simplified. Specifically, we reformulate it as environment lighting estimation at a designated location, conditioned on the reconstructed 3D Gaussian radiance field. Our solution proceeds in three steps:

- **Panoramic Projection.** The reconstructed radiance field is projected onto a panoramic sphere centered at the object placement location. This is achieved through $360°$ ray tracing from the location, producing a partial local environment map comprising an RGB image, a normal map, and an alpha mask that distinguishes reconstructed from missing regions.
- **Panorama Completion.** To complete the partial panoramas, we fine-tune Stable Diffusion 2.1 (Rombach et al., 2022). In addition to the RGB images and mask channels, the corresponding normal maps are also fed into the network to provide geometric guidance during inference, which yields slightly improved performance, as shown in Table 6.
- **HDR Expansion.** Finally, the completed panoramas are extended to high dynamic range. We adopt an EV-conditioned prompting strategy (Phongthawee et al., 2024), fine-tuning the model to generate LDR panoramas at exposure values of –5, –2.5, and 0. These outputs are subsequently fused into a single, comprehensive HDR environment map.

### D.2  TRAINING DATASET

As discussed in Section D.1, the main challenge of our task lies in *Panorama Completion*. Compared with conventional natural image completion, our task has two distinctive characteristics:

- **Panoramic input**: the input images are panoramas rather than ordinary perspective images.

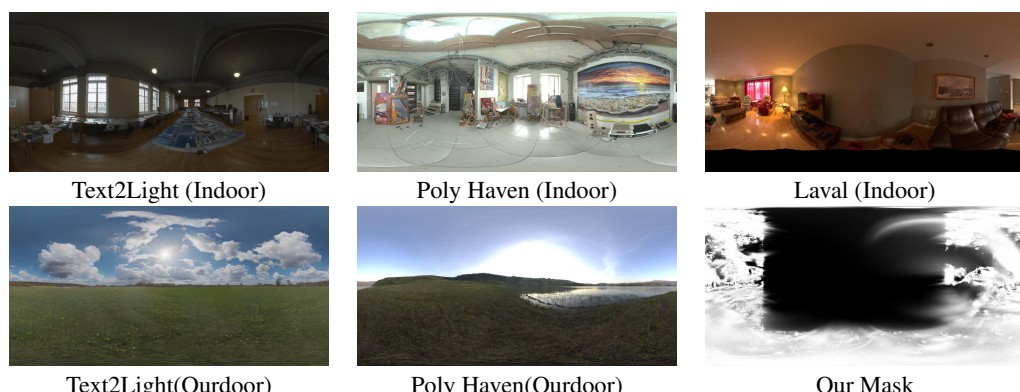

Text2Light (Indoor)    Poly Haven (Indoor)    Laval (Indoor)

Text2Light(Ourdoor)    Poly Haven(Ourdoor)    Our Mask

Figure 21: **HDR environment maps and a Gaussian mask from our training dataset.** We leverage publicly available environment maps from PolyHaven and synthesize a large, diverse set of HDR environment maps using the text-to-light model, Text2Light (Chen et al., 2022). Gaussian masks are generated from publicly reconstructed Gaussian scenes and exhibit artifacts typical of Gaussian radiance fields, such as *needle artifacts*, which are similar to those encountered in practical applications. This carefully curated dataset is well suited for training our panorama completion model.

- **Gaussian-derived masks**: the masks are traced from the Gaussian scene and exhibit unique *Gaussian characteristics*, such as *needle-like artifacts*.

To better adapt the network to this setting, our fine-tuning step relies on two key resources: (1) a sufficiently large and diverse collection of panoramic RGB images, and (2) a substantial set of masks generated from Gaussian scenes. Training samples are constructed by pairing each RGB image with a mask. Figure 21 illustrates representative panoramas and masks from our dataset. In the following, we provide a detailed account of how these data are obtained.

**HDR environment maps** Following DiffusionLight (Phongthawee et al., 2024), we used Text2Light (Chen et al., 2022) to generate numerous HDR panoramas from text descriptions. We use Large Language Models (LLMs) to generate textual descriptions. To ensure diversity and minimize prompt similarity, we adopt *a multi-LLM strategy*. Specifically, we employ three distinct LLMs: ChatGPT[4] , Grok[5] , and Gemini[6] . Each model produces 500 descriptions for indoor scenes and 500 for outdoor scenes. In addition, we incorporate HDR environment maps from public repositories such as Polyhaven and the Laval Indoor HDR (Gardner et al., 2017).

After curation and removing corrupted files, our final collection totals 5991 HDR environment maps, with 3938 indoor and 2053 outdoor scenes. We then randomly partitioned this collection into training and valid sets following an 8:2 ratio, resulting in 4792 and 1199 maps, respectively. For each map, we further obtain corresponding normal maps by using an off-the-shelf StableNormal (Ye et al., 2024a) model to infer these cues.

**Mask from Gaussian Splatting Scenes** We process various publicly available scenes (Barron et al., 2022) to obtain their 2D Gaussian representations and randomly select multiple points within them. From each point, we perform a 360° panoramic trace of the scene's radiance field. We then retain only the resulting alpha maps as masks for our dataset construction.

Assuming the scenes are Manhattan-aligned with the Z-axis pointing upwards, panoramic views of reconstructed Gaussian scenes often exhibit missing data in the upper hemisphere. To make our training data more representative of this scenario and to emphasize Gaussian artifacts, we apply a *random-region dropout* strategy during tracing. Specifically, before tracing, we exclude Gaussian primitives within a randomly oriented solid angle from the trace point. This generates sharper and more challenging mask boundaries, better reflecting real-world conditions.

---

[4] https://chat.openai.com/
[5] https://x.ai/
[6] https://deepmind.google/models/gemini/

## D.3 TRAINING DETAILS

Our Lighting Estimation model is fine-tuned from the pre-trained Stable Diffusion 2.1 (Rombach et al., 2022). We employ an EV-conditioned prompting strategy (Phongthawee et al., 2024) to generate environment map completions corresponding to specified Exposure Values (EVs). Finally, the outputs generated under different EVs (e.g., -5, -2.5, 0) are fused into a single, comprehensive HDR environment map. Our training process is divided into two main steps. The first step aims to adapt the model for LDR panorama completion. Then the second step further fine-tunes the model to be conditioned on EV prompts, enabling direct control over the exposure of the generated results.

**Step 1: Fine-tuning on LDR Panorama Completion.** In the first step, the model input is formed by concatenating the partial RGB image, its corresponding normal map and alpha mask, at a resolution of $512 \times 1024$ pixels. To guide the completion process, we use a fixed prompt: "*A realistic 360 degree panoramic [indoor/outdoor] scene, overexposed, bright*". We manually specify the indoor or outdoor token based on the scene type to help the model better distinguish and adapt to the significant content disparities between indoor (e.g., ceilings and lights) and outdoor (e.g., sky and sun) scenes, particularly in the upper regions of the panorama.

To enhance the model's generalization and robustness, we apply two data augmentation techniques to the input HDR data: random rotation and random EV adjustment. Note that the EV adjustment here serves only as data augmentation. The model is trained to produce an output with the same EV as the augmented input, with no explicit EV conditioning. We train this step by tasking the model with a v-prediction objective, minimizing an L1 loss between the prediction and the ground truth.

**Step 2: EV-Conditioned Fine-tuning** The core of the second step is the introduction of EV-conditioned prompting strategy. We dynamically generate prompts for arbitrary EV by linearly interpolating the text embeddings of two base prompts. Specifically, we define a base prompt embedding $\xi_b$ for EV=0 (the prompt used in step 1) and a dark prompt embedding $\xi_d$ for $EV_{\min} = -5$ (by replacing "*overexposed, bright*" with "*underexposed, dark*"). The target prompt embedding for a given EV is then computed as $\xi_{ev} = \xi_b + (ev/EV_{\min})(\xi_d - \xi_b)$.

To create the training pairs, we first take an original HDR environment map $I_{\text{org}}$ and apply a random EV shift to it, resulting in an augmented map $I_{\text{aug}}$ with $EV_{aug}$. Then, we apply a second, randomly sampled conditional exposure value, denoted as $EV_{\text{gt}}$, and generate the corresponding ground truth image $I_{\text{gt}}$ for supervision. The model is trained to adjust the exposure from the input $EV_{aug}$ to the target $EV_{\text{gt}}$. All other training settings remain the same as in step 1.

**Implementation Details** We train our model using the AdamW optimizer with $\beta_1 = 0.9$, $\beta_2 = 0.999$, and a weight decay of 0.01. The learning rate follows a constant schedule with a linear warmup of 500 steps. We train step 1 for 65 epochs with a learning rate of $1 \times 10^{-4}$ and fine-tune step 2 for 20 epochs with a reduced learning rate of $5 \times 10^{-6}$. The batch size is 8 for both steps. All models are implemented in PyTorch and run on NVIDIA A6000 (48GB) GPUs. step 1 training uses 4 GPUs and takes approximately 6.5 hours, while step 2 runs on 2 GPUs for about 3 hours.

## D.4 ABLATIONS

We conduct ablation studies on our lighting estimation method. First, we investigate the impact of scene completeness, measured by the effective area of the mask, on lighting estimation. Second, we examine the effect of including surface normal information as an additional input channel. We adopt the widely used evaluation protocol with an array of spheres from prior lighting estimation works (Gardner et al., 2017; 2019). The evaluation metrics include scale-invariant Root Mean Square Error (si-RMSE), Angular Error, and normalized RMSE.

**Ablations on Scene Completeness.** Following the method described in Section D.2, we generate 5,000 masks and group them according to the proportion of their valid coverage area. Specifically, the masks are divided into three groups: 40–60%, 60–80%, and 80–100%, and the statistics of each group are analyzed. Masks with less than 40% coverage are excluded, as such cases typically correspond to insufficient scene images or suboptimal object placement, making them unlikely to reflect practical usage. We then evaluate the model's performance across these three groups using the protocols described above. The results in Table 6 show that the accuracy of lighting estimation

Table 6: **Ablations on Lighting Estimation.** Performance improves with higher mask coverage, and incorporating the normal map enhances lighting estimation by providing explicit geometric cues.

| | Scale-invariant RMSE ↓ | Angular Error ↓ | Normalized RMSE ↓ |
|---|---|---|---|
| coverage 40-60% | 0.066 | 6.037 | 0.103 |
| coverage 60-80% | 0.060 | 5.599 | 0.089 |
| coverage 80-100% | **0.052** | **4.967** | **0.074** |
| w/o Normals | 0.067 | 0.061 | 0.100 |
| with Normals | **0.065** | **0.058** | **0.098** |

consistently increases with higher effective coverage. This trend is expected, since larger coverage provides more informative scene data for the model to infer illumination. These findings suggest that, in practical usage, capturing the scene as completely as possible is beneficial for achieving more reliable and accurate lighting estimation.

**Ablations on Geometric Information.** We compare two model settings: our full model, where the UNet input is a concatenation of the RGB image, normal map, and alpha mask; and an ablated version (w/o Normal Map), where the normal map is omitted from the input. To ensure a fair comparison, both models were trained from scratch using identical training data and hyperparameters. Their performance is then evaluated using the same protocols and metrics. As shown in Table 4, incorporating the normal map leads to more accurate lighting estimation, which can be attributed to the additional structural information provided by the normals.

## E    LLM Usage Statement

We use ChatGPT, a large language model developed by OpenAI, to polish the language of our manuscript. The model is not involved in research ideation, experimental design, data analysis, or the drawing of conclusions. The authors take full responsibility for the content of the paper.

