# OpenReview forum: "ComGS: Efficient 3D Object-Scene Composition via Surface Octahedral Probes"
_ICLR.cc/2026/Conference — ICLR 2026 Poster_

### Official Review · Reviewer_4c68 · 2025-10-25

**Soundness:** 3
**Presentation:** 3
**Contribution:** 3
**Rating:** 6
**Confidence:** 5

**Summary:**

This paper proposes ComGS, a Gaussian-splatting-based framework that performs inverse rendering for a scene and supports inserting new objects into the scene with consistent lighting appearance. Main contributions include (1) Surface Octahedral Probes (SOPs) for capturing indirect lighting and occlusion, (2) a diffusion model to estimate HDR 360° lighting based on the Gaussian scene, (3) the occlusion caching technique to efficiently cast shadows of the new object without ray tracing. Experimental results demonstrate superior quality to baseline.

**Strengths:**

1. The idea of using light probes for the inverse rendering task is novel. Light probes are a technique in traditional graphics rendering that cache indirect lighting to avoid costly ray tracing, which is well-suited to the inverse rendering task for performance gains.
2. The proposed occlusion caching further avoids costly ray tracing for the inserted object. Ray tracing has been a widely used but costly technique in recent years' inverse rendering paper, and this paper successfully bypasses it while preserving quality.
3. The inverse rendering community can benefit from the SynCom Dataset created by this paper.

**Weaknesses:**

## Major
1. Missing citations. The related work section only introduces NeRF- and GS-based inverse rendering methods, while a rich literature of mesh-based inverse rendering methods has also emerged in recent years and is worth mentioning, for example,
```
@article{hasselgren2022shape,
  title={Shape, light, and material decomposition from images using monte carlo rendering and denoising},
  author={Hasselgren, Jon and Hofmann, Nikolai and Munkberg, Jacob},
  journal={Advances in Neural Information Processing Systems},
  volume={35},
  pages={22856--22869},
  year={2022}
}
@inproceedings{dai2025inverse,
  title={Inverse Rendering using Multi-Bounce Path Tracing and Reservoir Sampling},
  author={Dai, Yuxin and Wang, Qi and Zhu, Jingsen and Xi, Dianbing and Huo, Yuchi and Qian, Chen and He, Ying},
  booktitle={The Thirteenth International Conference on Learning Representations},
  year = {2025},
  url = {https://openreview.net/forum?id=KEXoZxTwbr}
}
```
The idea of using a diffusion model to estimate 360° environment map lighting is also explored by previous work
```
@article{lyu2023diffusion,
  title={Diffusion posterior illumination for ambiguity-aware inverse rendering},
  author={Lyu, Linjie and Tewari, Ayush and Habermann, Marc and Saito, Shunsuke and Zollh{\"o}fer, Michael and Leimk{\"u}hler, Thomas and Theobalt, Christian},
  journal={ACM Transactions on Graphics (TOG)},
  volume={42},
  number={6},
  pages={1--14},
  year={2023},
  publisher={ACM New York, NY, USA}
}
```
2. Lambertian assumption. The training scheme regularizes the roughness to be near 1 and metallic to be near 0 (Eq. 12). While this may improve training stability, it also limits the expressivity of the inverse rendering pipeline, especially for scenes with specular reflections. Also, Eq. 15 further approximates the rendering equation with Lambertian assumption, which may introduce bias for non-Lambertian scenes.
3. Paper writing. (1) The proposed Surface Octahedral Probes are one of the main contributions of this paper, but I found the corresponding description to be too concise (L224-235). The authors should expand the corresponding paragraph in the revision. (2) The paper does not contain a limitation subsection.

## Minor
1. Monte Carlo rendering. Strictly speaking, the rendering estimator in Eq. 7 with a low-discrepancy sequence should be a deterministic quasi-Monte-Carlo scheme, compared to traditional Monte Carlo methods that require stochastic importance sampling. Despite avoiding noise and variance, the shortcoming of such a scheme is the lack of ability to capture high-frequency details, such as sharp specular reflections [1]. This weakness coincides with the Lambertian assumption weakness I mentioned before.
2. The inverse rendering results on TensoIR Dataset (Fig 14) still contain significant artifacts, e.g. the albedo map. It would also be preferred to show the inverse rendering results on the scene-level data.

[1] Zhu et al. "Learning-based inverse rendering of complex indoor scenes with differentiable Monte Carlo raytracing." SIGGRAPH Asia 2022 Conference Papers. 2022. Fig 4

**Questions:**

1. Is the proposed 360° environment map lighting diffusion model (Sec 3.2.1) similar to the "Diffusion posterior illumination" paper? If so, the author should clarify this and weaken the contribution claim on it.
2. Can authors provide a clarification on whether the method can handle non-Lambertian scenes? Experimental results are also preferred.
3. How accurate are the estimated cast shadows by the occlusion caching compared to the ray-traced ground truths, given that Eq. 15 may introduce significant approximation error?

---

> ### Author Response · Authors · 2025-11-28
> **Reply to Reviewer 4c68, Part 1**
>
> We appreciate your professional insights from the perspective of computer graphics. Please find our point-by-point responses below.
>
> ### **Major Weakness 1 and Question 1: Missing citations and Similarity with "Diffusion posterior illumination".**
>
> We thank the reviewer for pointing out these important and highly relevant references. We have now added the suggested citations to our related work section.
> We have carefully reviewed the paper "Diffusion posterior illumination"[1] and found its approach very interesting, especially the integration of differentiable rendering with a denoising process for illumination estimation and material decomposition.
>
> We note that **our method differs from theirs in several key aspects**. Our approach is conceptually simpler and builds more directly upon DiffusionLight [2]. Specifically, while DiffusionLight estimates an HDR environment map from a single RGB image but struggles with view consistency, we propose to use the pre-reconstructed radiance field of the scene to infer lighting. Moreover, in our work, lighting estimation is not positioned as a primary contribution, but rather serves as a necessary component within our overall composition pipeline.
>
> [1] Linjie Lyu, Ayush Tewari, Marc Habermann, Shunsuke Saito, Michael Zollhöfer, Thomas Leimkühler, and Christian Theobalt. Diffusion posterior illumination for ambiguity-aware inverse rendering. ToG, 2023.
>
> [2] Pakkapon Phongthawee, Worameth Chinchuthakun, Nontaphat Sinsunthithet, Varun Jampani, Amit Raj, Pramook Khungurn, and Supasorn Suwajanakorn. Diffusionlight:Light probes for free by painting a chrome ball. In CVPR, 2024.
>
> ### **Major Weakness 2: Lambertian assumption.**
>
> First, regarding scene reconstruction, we would like to clarify that our method does not estimate material parameters (such as albedo, roughness, or metallic) for the scene. As noted in the manuscript, accurate lighting estimation for complex real-world scenes remains highly challenging. Both Gaussian-based and learning-based inverse rendering methods often produce unreliable lighting due to complex light interactions and potentially incomplete image coverage, making the problem ill-posed. Material parameters estimated under such lighting are also likely to be inaccurate.
>
> To make the problem of 3D object-scene composition more tractable, we intentionally focus only on reconstructed scene geometry and the radiance field, and base lighting estimation on this representation.
>
> Second, we acknowledge that for the underlying 2DGS reconstruction method, geometry estimation itself is challenging in highly specular scenes, and the quality of geometry is crucial for subsequent lighting estimation and material decomposition.
>
> Finally, the Lambertian assumption used in Eq. 15 is a deliberate simplification to make the challenging problem of 3D realistic composition more manageable.
>
> ### **Major Weakness 3: Concise description of SOPs.**
>
> Thank you for your valuable suggestion. We agree that providing a more comprehensive explanation of the SOPs will help highlight their contribution. The concept of SOPs is a central thread that runs throughout the paper. The initial description in the manuscript was intended to introduce the core idea, with their application detailed in subsequent sections. For the reviewer's convenience, the key passages discussing SOPs are:
>
> - L224-L228: Overview of SOPs
> - L228-L235: Strategy for SOPs Placement
> - L236-L254: Querying Lighting and Visibility with SOPs
> - L260-L262: Supervising SOPs' Textures via Tracing
> - L291-L312: Using SOPs for Occlusion Caching
> - L338-L344: Shadow Calculation using SOPs
> - L514-L524: Discussions on SOPs placement and updates
>
> In the revised manuscript, we have expanded cotents related to SOPs for a clearer and more detailed representation.
>
> ### **Minor Weakness 1: Monte Carlo rendering.**
>
> You are correct in noting that using a low-discrepancy sequence results in a deterministic quasi-Monte Carlo scheme.
>
> In our implementation, while we employ such a sequence for sampling, we follow the practice established in prior inverse rendering researches by applying a random rotation to the rays at each iteration during reconstruction. This helps to mitigate potential issues.
>
> Furthermore, during the final rendering stage, we utilize importance sampling based on the octahedral texture, which is better suited for handling high-frequency details.
>
> We apologize for the lack of clarity regarding these implementation details in the original manuscript, and we have addressed this in the revised version.

---

> ### Author Response · Authors · 2025-11-28
> **Reply to Reviewer 4c68, Part 2**
>
> ### **Minor Weakness 2: Scene-level Inverse Rendering.**
>
> Thank you for this comment. Our work focuses on the challenge of realistic 3D object-scene composition. Consequently, our pipeline is designed to prioritize lighting estimation on scene, which is crucial for composition, rather than solving for full scene-level material properties. This is why we adopted a diffusion-based approach after finding that inverse rendering methods often struggle with lighting quality in complex scenes.
>
> Therefore, while we appreciate the value of high-quality inverse rendering, it falls outside the scope of our paper's main contribution. The effectiveness of our method is demonstrated by the final composition results (**Table 1, Figure 7, Figure 11 and Figure 14**), and we do not claim superior scene-level material decomposition, as that is not our objective.
>
> ### **Question 2: Performance on non-Lambertian scenes.**
>
> Our method has a limitation in handling strongly non-Lambertian surfaces, such as those with sharp specular reflections. We verified this through an experiment on the SynCom dataset, where we modified the table material in the "Room" scene to be highly metallic. The results were suboptimal, which we attribute to two main factors. First, the accurate reconstruction of reflective surfaces remains a challenging problem for the underlying scene representation. Second, and more critically for our pipeline, the current method cannot infer that a surface is highly reflective, thus it cannot render complex effects like mirror reflections correctly.
>
> We regard this as an important direction for future work. Potential solutions may include integrating robust material priors or leveraging diffusion models to render such advanced light transport effects. A detailed discussion of this limitation and potential solutions has been included in the new "DISCUSSION" section of the revised manuscript.
>
> ### **Question 3: Effectiveness and Limitations of the Occlusion Caching Method.**
>
> We acknowledge that the estimated shadows are approximations. The problem of achieving perfect physical accuracy for 3D object composition in arbitrary scenes is highly challenging, primarily due to its inherent ill-posed nature. The approximation in Eq. 15 is a deliberate simplification we introduced to make this complex problem tractable.
>
> While our results do not match perfectly the physical precision of the ray-traced ground truths from the Blender Cycles engine (as shown in **Figure 7**), they demonstrate a significant improvement over existing baseline methods. Our approach generates shadows that are visually coherent with the scene lighting.
>
> Handling more challenging scenes, such as highly specular scenes, is a valuable direction for future work. We have discussed this limitation and the rationale behind our design choices in the newly added "DISCUSSION" section.

---

### Official Review · Reviewer_5io7 · 2025-10-29

**Soundness:** 2
**Presentation:** 3
**Contribution:** 2
**Rating:** 6
**Confidence:** 4

**Summary:**

This paper presents a novel object-scene composition framework, ComGS, which consists of an efficient 2DGS-based inverse rendering method and a diffusion-based lighting estimation model. The proposed inverse rendering pipeline utilizes Surface Octahedral Probes to cache the indirect illumination and occlusion, eliminating the need for costly ray tracing . Experiment results are provided to demonstrate the good performance of the framework, highlighting the effectiveness of SOPs.

**Strengths:**

1. This paper proposes a novel inverse rendering framework. Unlike previous methods that use expensive ray tracing to obtain indirect lighting and occlusion, this method uses probes to store indirect lighting and occlusion, significantly improving efficiency.
2. The proposed lighting estimation model conditioned on rendered partial panorama envmap from GS demonstrates good performance.
3. Extensive experiments show that the method achieve better performance than inverse-rendering and generative baselines.

**Weaknesses:**

1. It seems that the proposed methods cannot be used in single image scenario. During editing stage, since the object is inserted into the scene, the occlusion probes near the object should be baked first.  But if the scene is represented by single image, it is impossible to use GS to reconstruct it, which means we cannot obtain the occlusion SOPs.
2. Even the position of probes are generated from the surface point clouds using FPS, the sampled probes may still be within the Gaussians’ effective ranges (mixing with the Gaussians), which will cause light leakage. But I think there shouldn't be many such examples, so this isn't a big problem.

**Questions:**

1. Is there a way to obtain occlusion probes even when the scene is represented by a single image? Or are there other methods to obtain the shadow?
2. Could you provide the performance of two extra baselines MV-CoLight and GI-GS on your SynCom-Objectdataset?

[MV-CoLight, NIPS'25](https://github.com/InternRobotics/MV-CoLight)

[GI-GS, ICLR'25](https://github.com/stopaimme/GI-GS/tree/master)

---

> ### Author Response · Authors · 2025-11-28
> **Reply to Reviewer 5io7**
>
> We thank you for your time and valuable comments. Our responses to each point are as follows.
>
> ### **Weakness 1 and Question 1: Single Image Scenario.**
>
> You are correct that our method requires a multi-view reconstructed 3D scene, which is not available from a single image.
>
> One way to extend our method to single-image inputs would be to first generate a 3D Gaussian representation using a feed-forward network and then apply our pipeline to this generated scene. However, comprehensively addressing the single-image case might present challenges that are beyond the scope of this paper. Our work focuses on 3D object-scene composition starting from multiple input views, performing reconstruction, editing, and rendering in a unified process.
>
> We have added a discussion in **revised Section 5** to note the single-image scenario as a promising direction for future work.
>
> ### **Weakness 2: Possible light leakage.**
>
> We agree that careful probe placement is essential.
>
> In our implementation, we mitigate this issue by introducing a small offset. Specifically, after sampling probe positions using FPS, we shift each one outward along the surface normal by a distance equivalent to 1% of the object's size. This helps to distance the probes from the core density areas of the Gaussians.
>
> We conducted an ablation study on the offset value using the *Toy* scene from the SynCom dataset. The results are as follows:
>
> | Offset | Albedo PSNR | Rendering PSNR | Relight PSNR |
> |---|---:|---:|---:|
> | 0% | 33.222 | 37.204 | 30.449 |
> | 1% | **33.304** | **37.265** | **31.324** |
> | 2% | 33.130 | 37.202 | 31.281 |
>
> The 1% offset provided the best overall performance across all metrics. This setting was adopted for all experiments in the paper, and we did not observe significant light leakage in our results.
>
> We believe that exploring more advanced, potentially automatic, probe placement strategies is an interesting direction for future work. We have added a brief discussion on this point in the **revised Section 5**.
>
> ### **Question 2: Two extra baselines.**
>
> Thank you for the suggestion. We have evaluated the two baselines, MV-CoLight and GI-GS, as requested.
>
> To clarify the experimental setup, the composition task involves inserting objects from the SynCom-Object collection into scenes from the SynCom-Scene collection, and performing​ evaluation on SynCom-Composition collection, thus utilizing the full SynCom dataset. The performance of all methods, including these new baselines, has been evaluated in **Table 1**.
>
> Importantly, to ensure a fair and consistent subjective comparison, we have re-conducted the subjective evaluation for all methods. This is essential to control for potential confounding factors, such as variations in the timing of experiments or the participant pool, which could otherwise introduce bias.

---

### Official Review · Reviewer_safG · 2025-10-30

**Soundness:** 4
**Presentation:** 4
**Contribution:** 3
**Rating:** 6
**Confidence:** 4

**Summary:**

This paper presents a realistic 3D object-scene composition pipeline that achieves plausible shadows and lighting effects in real time.
The framework consists of three major stages:
(1) Reconstruction, where both the object and scene are represented using Gaussian splatting;
(2) Editing, which includes environment lighting estimation and Surface Octahedral Probe (SOP)-based occlusion caching; and
(3) Rendering, performing relighting and shadow synthesis for seamless composition.
The proposed system is efficient and demonstrates high-quality visual realism on various examples of single object insertion and relighting.

**Strengths:**

1. The proposed framework demonstrates good performance and maintains efficiency, outperforming several baselines.
2. The proposed SOP structure is convenient, enabling real-time shadow and lighting computation.
3. The paper is well organized and easy to follow.

**Weaknesses:**

1. Self-occlusion or ambient occlusion effects on the inserted object are barely noticeable (e.g., Figure 7), suggesting that the occlusion modeling may not be effective enough. (Please also see question 3).
2. The experiments mostly showcase single-object insertions. It remains unclear how the system performs when inserting multiple objects, or when an object is placed freely in the scene (for example, partially under the shadow area). More results could further demonstrate the effectiveness of the proposed method.
3. The performance of object inverse rendering depends on the input images. The synthetic dataset from the simulator largely aligns with the proposed method, which may be biased. More discussion about real-data capture conditions or constraints would improve clarity and generalizability.

**Questions:**

1. For the scene reconstruction, are the 2D Gaussians also equipped with albedo, roughness, and metallic attributes as described in lines 155–161? If so, how are these optimized since they do not appear in Step 2? For the object reconstruction, is the RGB buffer C still optimized and used? If so, how does it involve in the rendering equation?
2. Eq. 2 seems incorrect, any typos?
3. Is the self-occlusion term computed in Eq. (8) reused in Section 3.3 (``Rendering'')? Does the relighting step re-bake indirect illumination under new lighting conditions? From Figure 7 and explanations in Section 3.3, it seems these two terms are not re-used at the relighting stage.
4. What if there are multiple light sources in the scene (may correspond to various peak values in the environment map)? Will the generated shadow composite of layered shadows with darker overlapped regions?
5. Does a mismatch between the captured object’s original environment map and the target scene environment influence the final performance?

I will raise the score if the major concerns are addressed.

---

> ### Author Response · Authors · 2025-11-28
> **Reply to Reviewer safG, Part 1**
>
> We are grateful for your highly constructive comments, which have been very helpful. Our responses are detailed below:
>
> ### **Weakness 1 and Question 3: Barely noticeable self-occlusion.**
>
> We sincerely thank the reviewer for this insightful comment and for pointing out the lack of noticeable self-occlusion effects. This is a valuable observation that has helped us improve our work.
>
> In our initial submission, the self-occlusion modeled by the Surface Octahedral Probes (SOPs) was indeed not applied during the final composition stage, which led to less realistic results as correctly noted. Following your suggestion, we have now integrated the SOPs into our composition pipeline to account for self-occlusion. Specifically, we modify the rendering equation (Equation 8) to modulate the direct lighting $L_{dir}(\omega_i)$ by the visibility factor $(1 - O(\omega_i))$ from the SOPs, resulting in
>
> $L_i(\omega_i) = (1 - O(\omega_i)) L_{dir}(\omega_i).$
>
> This is then used in the Monte Carlo integration (Equation 7) to render the re-lit object with self-occlusion before composition.
>
> We will update all relevant experiments gradually. The updated results in Table 1 and Figure 7 demonstrate that the rendered objects now exhibit more convincing self-shadowing, enhancing the overall realism. We have also revised Section 3.3 to provide a detailed explanation. In Figure 16, we present a comparison between results with and without self-occlusion applied.
>
> For indirect lighting, which demands a comprehensive global illumination model, we made a conscious decision to exclude it due to the prohibitive computational cost.
>
> We believe the revised manuscript is substantially improved thanks to this comment.
>
> ### **Weakness 2: Multi-object insertions.**
>
> To address the concern, we have conducted additional experiments involving the insertion of two objects, where one object is moved to different locations in the scene. These results are included in Figure 17 of the revised manuscript.
>
> Although our method does not directly support inserting multiple objects simultaneously, we are able to achieve multi-object insertion **in a sequential manner**: after inserting the first object, the modified scene is treated as a new base scene. We render multi-view images of this new scene and use Stage 1 to reconstruct it. The insertion of a second object then becomes an object insertion problem into this updated scene. Our experiments show that our approach can successfully handle multiple object insertions, and can also model partial occlusion between objects.
>
> ### **Weakness 3: Synthetic Data and Generalizability.**
>
> Our primary objective is photorealistic 3D object-scene composition, rather than solving the general inverse rendering problem. The choice of synthetic data is motivated by the need for controlled conditions, as they​ provide **precise control over object placement and camera settings**. This control is crucial for establishing a reliable benchmark for 3D composition and facilitating quantitative assessment. Our object rendering follows the traditional environment map-based setting adopted by previous inverse rendering works, and the synthetic scenes are manually adjusted to approximate real-world conditions as closely as possible.
>
> We fully agree that validation on real data is essential. To that end, we have extensively tested our method on **public datasets (See Figure 11)** and **real-world captures (See Figures 14)**. In **Section B.2** of the revised manuscript, we describe our data collection process: objects were placed under soft lighting and captured using a systematic spiral camera trajectory to ensure robust viewpoint coverage. Furthermore, in **Figure 13**, we have included sample images and camera pose visualizations from these real-world sequences to better illustrate the practical conditions and enhance the discussion on generalizability.

---

> ### Author Response · Authors · 2025-11-28
> **Reply to Reviewer safG, Part 2**
>
> ### **Question 1: Details on Scene and Object Reconstruction.**
>
> We do not​ estimate material parameters (albedo, roughness, metallic) for the scene. As discussed in the manuscript, accurate lighting estimation for complex real-world scenes remains a significant challenge. Both Gaussian-based inverse rendering methods and learning-based estimation techniques​ often struggle to recover satisfactory lighting results. This is due to the complex light interactions and potentially incomplete scene coverage in the input images, which lead to an ill-posed problem. Consequently, the material parameters estimated under such unreliable lighting are also unlikely to be accurate.
>
> To make the problem more tractable, we deliberately focus only on reconstructing the scene's geometry and radiance field. We then base our lighting estimation on this representation.
>
> Regarding object-level reconstruction: The RGB buffer $\mathcal{C}$, obtained via splatting of SH-based colors $\mathbf{c}_i$, is used and optimized only during the first stage​ (radiance and geometry reconstruction). It is supervised by a photometric loss in Eq. (2). In the second stage, this SH-based RGB buffer is no longer optimized. Instead, we use a deferred PBR shading model to produce a physically-based image, which is then supervised by a loss function formulated similarly to Eq. (2).
>
>
> ### **Question 2: Possible typos in Eq.2.**
>
> After careful review, we don't find typos in Eq. (2). The impression of a possible issue may have arisen from our initial reference to Eq. (2) in the context of PBR supervision. To prevent any misunderstanding, we have revised the relevant sentence in the manuscript from:
>
> "We use the render loss $\mathcal{L}_{pbr}$ for PBR as in Eq. (2)."
>
> to the more precise formulation:
>
> "We use a rendering loss similar to that in Eq. (2) for the PBR image $\mathcal{C}_{pbr}$, denoted as $\mathcal{L}_{pbr}$."
>
> We hope this clarification adequately addresses your concern.
>
> ### **Question 4: Shadow from multiple light sources.**
>
> Yes, our method can handle multiple light sources, and the generated shadows composite in a way that overlapping regions become darker. To validate this, we conducted an experiment on the 'Kettle_in_Room' scene by manually adding a secondary point light source to its environment map.
>
> After adding the new light source, the shadows cast by the original light become less pronounced due to the increase in overall illumination. Furthermore, in regions where shadows from both light sources overlap, a darker, composite shadow forms, as expected. We illustrate this result in **Figure 15** of the revised manuscript.
>
> ### **Question 5: Influence of Environment Mismatch.**
>
> There is indeed a mismatch between the lighting conditions during object capture and the target scenes. For example, in the synthetic dataset, objects are illuminated by the "Orbita" HDRI map, while the target scenes have complex global illumination (**Figure 10**). For real-captured objects, the mismatch also exists between capture area of coffee table and the target scenes like a hall or courtyard (**Figure 13**).
>
> Our method is designed to handle this expected mismatch. Since we model the object's relightable properties, it can adapt its appearance when placed into a new environment. Therefore, the performance of our method is not adversely affected by this difference in lighting conditions.

---

### Official Review · Reviewer_7Qbv · 2025-10-31

**Soundness:** 2
**Presentation:** 3
**Contribution:** 3
**Rating:** 6
**Confidence:** 3

**Summary:**

This work presents a novel pipeline for realistic 3D object-scene compositions, which is formulated as composing given scene and object multi-view images into a harmonious and visually-natural 3D representation. The main contributions of this work are surface octahedral probes (SOP) and local lighting completion. These designs empower high-quality relightable object reconstruction and object-scene composition with significantly improved efficiency.

**Strengths:**

1. The proposed method is well-motivated. This work targets two long-standing obstacles in the object-scene composition task: relightable object reconstruction and scene lighting estimation, and introduces specific designs to tackle these challenges.
2. The SOP design mitigates the reliance on per-point ray tracing during optimization and rendering, thus improving the inverse rendering efficiency, leading to (near) real-time relightable rendering speed.
3. The local lighting completion builds upon the insight that only part of the scene interacts with the to-be-composed object, thus focusing on only the salient parts of the representation.
4. The proposed pipeline achieves superior performance from both the quantitative and qualitative evaluations while demonstrating high rendering efficiency, verifying its optimal effectiveness-efficiency trade-off.

**Weaknesses:**

1. The SOP initialization raises concerns. It is assumed that the SOPs are initialized by ray tracing and then optimized in a supervised manner. This initial ray-tracing step and progressive SOPs require careful designs to avoid light leaks and placement offsets. Although a heuristic solution is provided for the offset = 1% object size, its robustness is not clearly verified.
2. Assumption on the pipeline settings. The proposed pipeline focuses on scenes with low or moderate occlusions (<40%), and builds on assumptions that the object is small relative to the scene and its placement affects only its own appearance and nearby regions. These assumptions are fundamental to the overall pipeline and prompt an efficient solution. However, it is not clear whether this assumption accounts for most real-world cases, and it is interesting to investigate how the method performs for hard cases and failure cases.

**Questions:**

1. It is recommended to add ablations or verifications on the SOP initializations. How the offset affects the final results and whether per-scene tuning is needed should be clarified.
2. The memory consumption introduced by SOPs should be shown, as thousands of probe operations are executed in this implementation.
3. How sensitive are results to SOP count and probe texture resolution? Is there a principled way to choose probe density per scene scale?
4. A deduction on the limit of the proposed method regarding the assumptions is favorable. For example, what is the threshold for a normal pipeline proceeding, and how do these assumptions affect the method’s effectiveness?
5. Is it possible to incrementally update the SOP as the object or camera moves to mitigate the calculation of re-caching for each editing? This may further enhance the pipeline completeness.

---

> ### Author Response · Authors · 2025-11-28
> **Reply to Reviewer 7Qbv, Part 1**
>
> We thank you for your time and thorough feedback. Our point-by-point responses are as follows:
>
> ### **Weakness 1 and Question 1: The SOP initialization raises concerns.**
> To evaluate the robustness of the heuristic offset setting, we conduct an additional ablation study by introducing different placement offsets of 0%, 1%, and 2% on the *Toy* scene in the SynCom dataset.
> | Offset | Albedo PSNR | Rendering PSNR | Relight PSNR |
> |---|---:|---:|---:|
> | 0 | 33.222 | 37.204 | 30.449 |
> | 1% | **33.304** | **37.265** | **31.324** |
> | 2% | 33.130 | 37.202 | 31.281 |
>
> The results show that **a 1% offset consistently achieves the best performance** in terms of material decomposition, rendering fidelity and relighting quality.
>
> In addition, we provide ambient occlusion and relighting visualizations under different offsets in **Figure 20** of the revised manuscript. As illustrated, the 0% setting introduces noticeable light leakage in both occlusion and relighting, while the 1% configuration yields clean and stable results. In all experiments reported in the paper, we adopt this unified 1% strategy, and we do not observe  notable failures caused by this choice.
>
> However, we agree that the 1% offset may not be a universally optimal principle, and scene-specific tuning could further improve performance. We plan to explore an adaptive SOP initialization strategy and develop a lightweight interactive tool for user-friendly adjustment, which we believe can further improve the overall usability of the pipeline.
>
> ### **Weakness 2 and Question 4.: Assumption on the pipeline settings.**
> Please see our **General Response**.
>
> ### **Question 2: The memory consumption introduced by SOPs.**
>
> We measure GPU memory usage on the *Bottom_in_Room* composition of the SynCom dataset under different numbers of SOPs, as summarized below:
>
> | Setting      | Memory (GB) |
> |--------------|------------:|
> | Ray Tracing  | 4.4         |
> | Num = 10,000 | 4.3         |
> | Num = 20,000 | 4.4         |
> | Num = 40,000 | 4.5         |
>
> The memory usage of 20K SOPs is comparable to that of a ray-tracing implementation (which requires maintaining a BVH structure). This indicates that the overhead of SOPs is moderate under typical settings. Although the memory consumption increases with the number of SOPs, this mild growth suggests that most of the memory is dominated by the Gaussian point cloud, which stores a large number of Gaussians with high-dimensional appearance attributes.
>
> In future work, we plan to explore adaptive SOP placement strategies to maintain the same visual quality with fewer probes, further reducing memory overhead.
>
> ### **Question 3: Sensitivity to SOP Count/Resolution.**
>
> In the revised manuscript, **Figure 15** demonstrates the impact of varying the number of SOPs and the probe texture resolution. Our baseline configuration uses 10k SOPs and a texture resolution of 16. We also tested configurations with lower (5k) and higher (20k) SOP counts, as well as texture resolutions of 8 and 32.
>
> When using only 5k SOPs or a texture resolution of 8, we observe noticeable shadow aliasing and a loss of fine shadow details. Specifically, at a resolution of 8, the angular sampling becomes too coarse over the full 360 degrees, leading to inadequate shadow representation and significant errors in shadow direction. A resolution of 16, however, produces acceptable results. Increasing either the SOP count or texture resolution reduces aliasing and improves the overall shadow quality.
>
> All our experiments are conducted using the default settings, and we did not observe severe failures. The results support a practical and robust guideline: begin with the baseline configuration (10k SOPs, resolution 16), which works well for a wide range of scenes. The SOP count can then be increased based on visual inspection to further reduce shadow artifacts if necessary.

---

> ### Author Response · Authors · 2025-11-28
> **Reply to Reviewer 7Qbv, Part 2**
>
> ### **Question 5: Incremental Update of SOPs.**
>
> SOPs are cached in **scene space**​ (not screen space), so they remain valid across camera moves and can be reused directly. However, when the inserted object moves, changes in visibility and lighting invalidate the cached SOPs. Our current pipeline can handle this by re-running the Editing Stage, which adds some computation but remains relatively fast.
>
> We agree that an incremental SOP update mechanism could further improve the pipeline. The core challenge lies in efficiently identifying which specific SOPs are affected by the object's movement and updating only those. Mesh-based methods, such as Dynamic Diffuse Global Illumination (DDGI)[1], leverage explicit surface geometry and acceleration structures to efficiently perform ray–surface intersections and incremental updates. However, Gaussian point clouds lack such structured representations, and ray–Gaussian intersection is more expensive, making efficient incremental updates substantially more challenging. Achieving this in full would necessitate further algorithmic design, which is beyond the current scope but represents an interesting avenue for future research.
>
> We clarify this challenge and discuss it in the newly added "DISCUSSION" section.
>
> [1] Zander Majercik, Jean-Philippe Guertin, Derek Nowrouzezahrai, and Morgan McGuire. Dynamic diffuse global illumination with ray-traced irradiance fields. JCGT, 2019.

---

### Author Response · Authors · 2025-11-28
**General Response**

We would like to express our sincere gratitude to the Area Chair and all reviewers for their insightful comments and constructive suggestions. It is a great encouragement to see our work recognized in several aspects, such as *the design of the SOPs for achieving real-time performance*, *the pipeline's ability to generate high-quality compositions*, and *the potential benefits of the created dataset*. We are inspired by these positive remarks to further improve our work. We have submitted a revised version of the manuscript, in which all modifications are highlighted in red.

Before providing point-by-point responses to each reviewer, we would like to first address some common concerns raised across the reviews. Specifically, these relate to certain underlying assumptions and methodological limitations of our paper, as highlighted in **Reviewer 7Qbv's Weakness #2** and **Question #4**, as well as **Reviewer 4c68's Major Weakness #2 and Question #3**. We provide a general response below.

### **Our Assumptions and Limitations**

We agree that a discussion on limitations is crucial, and we acknowledge that an initial lack of a dedicated section was an oversight. In response, we have added a new Section 5, "DISCUSSION" in the revised manuscript.

We acknowledge that the task of 3D object insertion into scenes is highly challenging, and existing methods struggle to achieve consistently satisfactory results. **To make the problem more tractable, we adopted certain assumptions that are also commonly used in prior work** [1, 2]. These include:

(1) The inserted object is relatively small, thereby ensuring no drastic alteration to the global illumination.

(2) The scene is predominantly Lambertian, allowing for a reasonable approximation of shadows cast by the inserted object.

We fully agree that these assumptions impose certain limitations on the applicability of our method. For instance, if the inserted object is very large or highly reflective to the extent that it significantly affects global illumination, our approach may produce unsatisfactory results. Similarly, in scenes with strong reflective surfaces, our method cannot model mirror-like reflections of the inserted object.

In the **newly added DISCUSSION section**, we summarize these limitations and present representative failure cases. Addressing these challenging scenarios remains an important direction for our future work.

[1] Z. Wang, W. Chen, D. Acuna, J. Kautz, and S. Fidler. Neural light field estimation for street scenes with differentiable virtual object insertion. In ECCV, 2022.

[2] Keyang Ye, Hongzhi Wu, Xin Tong, and Kun Zhou. A real-time method for inserting virtual objects into neural radiance fields. IEEE TVCG, 2024.

---

### Meta-Review · Area_Chair_Y2Ux · 2026-01-06

**Summary:**

Following is a summary of the reviewers' major concerns:

### Reviewer 7Qbv:
1. SOP initialization robustness.
    * **The authors acknowledge that the heuristic offset setting is not optimal.**
They provide additional experiment results on the Toy scene with different placement offsets to show the chosen parameter
gives the best performance.**
2. Assumptions on the scene are too restrictive. Not sure about its ability to handle complex scenarios with
complex occlusions.
    * **The authors clarify the assumptions and argue that these assumptions make the challenging problem more tractable.**

3. Efficiency and scalability of SOP.
    * **The authors added a table to show the memory cost of SOP**

### Reviewer safG:

1. Whether the method can handle self-occlusion.
    * **The authors added results with self-occlusion**.
2. Whether the method can support multi-object insertion.
    * **The authors reply that the proposed method can support multiple object insertions but would require
    reconstruction of the scene after each object insertion.**
3. Heavy reliance on synthetic data, need more real-world results.
    * **The authors added more results on real-world scenes.**


### Reviewer 5io7
1. Comparisons to extra baselines: MV-CoLight and GI-GS
    * **Comparisons are added.**


### Reviewer 4c68
1. Lambertian assumption on the object makes the paper lack the ability to handle
high-frequency specular effects.
    * ***The authors clarify the assumptions and argue that these assumptions make the challenging problem more tractable.**

**Reviewer Concerns:**

See above.

I think most concerns are well addressed in the rebuttal.  One common concern is that the paper makes very restrictive assumptions, but I think the author's argument on making the challenging problem more tractable is reasonable, and it does not affect the major technical contributions of the paper.

**Reviewer Scores:**

Reviewer 7Qbv: maintain the score

Reviewer safG: improve the score

Reviewer 5io7: maintain or improve the score

Reviewer 4c68: maintain

---

### Decision · Program_Chairs · 2026-01-26

Accept (Poster)